# Bi-Level Knowledge Transfer for Multi-Task Multi-Agent Reinforcement Learning

**Junkai Zhang**[1,2], **Jinmin He**[1,2], **Yifan Zhang**[1,2,3]*, **Yifan Zang**[4], **Ning Xu**[1], **Jian Cheng**[1,2,5]

[1] $C^2$DL, Institute of Automation, Chinese Academy of Sciences
[2] School of Artificial Intelligence, University of Chinese Academy of Sciences
[3] University of Chinese Academy of Sciences, Nanjing
[4] Beijing Institute of Astronautical Systems, [5] AiRiA
{zhangjunkai2021, hejinmin2021, zangyifan2019}@ia.ac.cn,
{yfzhang, jcheng}@nlpr.ia.ac.cn, nxu@njust.edu.cn

## Abstract

Multi-Agent Reinforcement Learning (MARL) has achieved remarkable success in various real-world scenarios, but its high cost of online training makes it impractical to learn each task from scratch. To enable effective policy reuse, we consider the problem of zero-shot generalization from offline data across multiple tasks. While prior work focuses on transferring individual skills of agents, we argue that the effective policy transfer across tasks should also capture the team-level coordination knowledge. In this paper, we propose **Bi**-Level **K**nowledge **T**ransfer (BiKT) for Multi-Task MARL, which performs knowledge transfer at both the individual and team levels. At the individual level, we extract transferable individual skill embeddings from offline MARL trajectories. At the team level, we define tactics as coordinated patterns of skill combinations and capture them by leveraging the learned skill embeddings. We map skill combinations into compact tactic embeddings and then construct a tactic codebook. To incorporate both skills and tactics into decision-making, we design a bi-level decision transformer that infers them in sequence. Our BiKT leverages both the generalizability of individual skills and the diversity of tactics, enabling the learned policy to perform effectively across multiple tasks. Extensive experiments on SMAC and MPE benchmarks demonstrate that BiKT achieves strong generalization to previously unseen tasks.

## 1 Introduction

Multi-Agent Reinforcement Learning (MARL) has shown great potential in a wide range of real-world applications, such as autonomous driving [4, 44], robotic collaboration [2, 11, 29], and distributed control [24, 39]. However, training MARL agents from scratch for each new task is often prohibitively expensive due to the high cost of online interactions and the complexity of coordination among agents[42, 45]. In real-world scenarios, we usually have access to offline MARL trajectories of some known tasks. Thus, distilling the multi-agent policy from such data and transferring it to unseen yet related tasks offers a cost-effective solution, which reduces reliance on online interaction. Consequently, we consider to solve the problem where agents are trained on known multi-tasks using offline data and then directly evaluated on previously unseen tasks, aiming to achieve zero-shot generalization.

---

*Corresponding author

39th Conference on Neural Information Processing Systems (NeurIPS 2025).

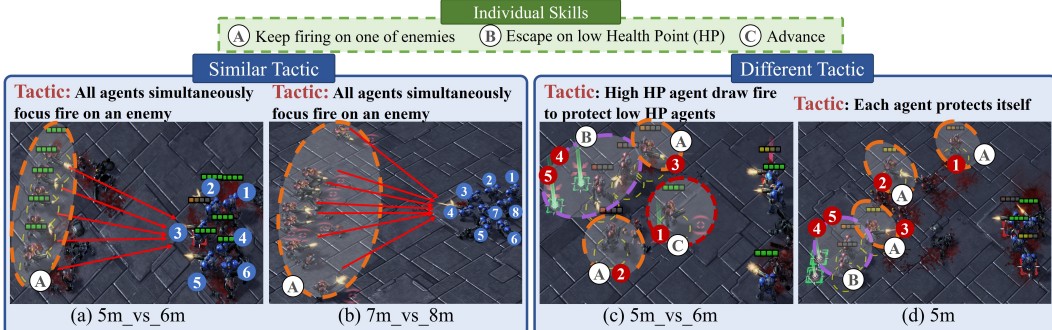

Figure 1: Skill and tactic examples in SMAC tasks. In (a) and (b), two tasks utilize a similar tactic: All agents keep fire on the same enemy, which forms the team tactic to quickly eliminate an enemy to gain a numerical advantage. In (c) and (d), two tasks utilize two different tactics: In (c), due to unbalanced HP. Agent 2,3 adopt Skill A, agents 4,5 use Skill B, and agent 1 adopts Skill C to draw enemy fire to protect the low-HP teammates. In (d), with balanced HP, agent 1,2,3 adopt Skill A and agent 4,5 adopt skill B, which corresponds to the tactic that each agent protects itself at low HPs.

An alternative approach is to leverage offline data to train a task-agnostic policy with a unified structure capable of taking actions in multiple tasks [9, 10]. However, they are primarily designed for online training, and their generalization heavily relies on the neural network's inherent capacity, which often fails when facing variations in agent number and corresponding observation spaces. Besides, the efficient policy transfer can also be achieved by the use of *skill*, which abstracts the action pattern of agents into compact representations [32]. This has been an effective strategy in single-agent generalization across tasks, allowing agents to reuse knowledge without learning from scratch. Inspired by it, recent studies have explored skill-based policy transfer to MARL [20, 41], where each agent learns to extract transferable skills and generates actions conditioned on them. We refer to these skills as *Individual skills*, which capture the policy of individual agents in MARL.

However, transferring individual skills alone is insufficient for effective policy transfer in MARL. While individual skills facilitate the knowledge transfer of individual action patterns, they fail to convey how these skills interact cooperatively within a team across tasks. Therefore, a comprehensive knowledge transfer in MARL should also capture the cooperative patterns of skills, which we define as *tactics*. As illustrated in Figure 1, different tasks may share similar tactics, which suggests that tactics can serve as valuable transferable knowledge. Besides, even in similar states, distinct tasks may require different tactical choices, which indicates the need for policy's adaptive selection in unseen tasks. Therefore, effective policy transfer in Multi-Task MARL requires two levels of knowledge: Skill transfer at the individual level and the tactic transfer at the team level.

In this paper, we propose our novel **Bi**-Level **K**nowledge **T**ransfer (BiKT) for Multi-Task Multi-Agent Reinforcement Learning. Specifically, at the individual level, we perform skill transfer by extracting latent skill representations from offline trajectories. At the team level, we leverage these learned skill embeddings to learn a compact tactic embedding, which captures the combinations of individual skills. We then discretize the tactic embeddings and organize them into a finite set, which we define as *tactic codebook*. To employ both the skills and tactics for decision-making, we design a bi-level decision transformer as the policy model, which selects appropriate tactics from the learned codebook and utilizes them to infer individual skills. The actions are then decoded from the skills. Through the above design, our BiKT benefits from the diversity of tactic codebook and the generalizability of individual skills, which can guide the policy to generalize in unseen tasks with high performance.

Our method exhibits the following contributions: 1) Our framework facilitates the bi-level knowledge transfer in multi-task MARL, enabling both individual and team-level generalization. 2) Our proposed tactic codebook captures the diversity of team behaviors across source tasks, providing the flexibility to generalize to a wider range of unseen tasks. 3) Our learned tactics serve as team-level guidance, enabling agents to select different skills under similar scenarios across tasks, which prevents the monotony of skill combinations. Extensive experiments in *Marine Hard*, *Marine Easy*, *Stalker Zealot*, and *Cooperative Navigation* in SMAC [31] and MPE [22] showcase our outcome performance.

## 2 Related Works

**Offline MARL** Offline reinforcement learning has emerged as a compelling paradigm for developing decision-making policies solely from static datasets [18]. Recent efforts have extended this paradigm to multi-agent settings, leading to the increasing prominence of offline MARL[6, 43]. A major challenge is the distributional shift between the training dataset and unseen state-action spaces, which results in inaccurate value predictions [7, 38]. To mitigate this, behavior-constrained learning frameworks have been proposed to encourage conservatism in policy updates [16, 17]. These conservative principles are often embedded within existing online MARL methods, either by extending multi-agent policy gradient methods [5, 22] or by applying value decomposition techniques for joint value estimation [30, 33, 35]. Furthermore, recent developments have concentrated on formulating safe and robust offline MARL algorithms, ensuring strong performance guarantees under distributional shifts [8, 26, 40]. However, excessive conservatism may hinder generalization, as policies trained on static data may overfit to the offline distribution. In parallel, diffusion-based generative models have been introduced to enhance offline policy learning [13, 27]. Despite their potential, integrating such models with the policy improvement remains a challenging problem [14, 36, 46].

**Multi-Task MARL** Multi-Task MARL aims to train a unified policy that can handle multiple tasks simultaneously and generalize directly to unseen tasks, which presents two primary challenges: the design of a universal policy and ensuring effective generalization across tasks. To enable a universal policy, flexible architectures empower universal policies to process variable inputs [1, 9, 10, 12, 47]. A curriculum learning strategy was grounded in evolutionary principles to enable scalability concerning the number of agents [21]. Methods such as randomized entity-wise factorization (REFIL) enhance policy adaptation by improving generalization [12], while transformer-based approaches like UPDeT leverage population-invariant networks to manage variable agent configurations [10, 19, 37]. Despite these advances, they usually rely on simultaneous learning or fine-tuning across tasks, which limits their generalization. To address this, the MATTER focuses on learning task representations to capture inter-agent relationships [28]. Additionally, ODIS [41] and HiSSD [20] promote generalization to novel tasks by incorporating skills within unified training paradigms. However, the pursuit of training policies robust enough for deployment in new tasks remains a key challenge, highlighting the need for continued research to push multi-task MARL towards more versatile solutions.

## 3 Preliminary

### 3.1 Cooperative Multi-Agent Reinforcement Learning.

A cooperative multi-agent task can be modeled as a Decentralized Partially Observable Markov Decision Process (Dec-POMDP) [25], which is defined by the tuple $(\mathcal{N}, S, \{o^i \in O\}_{i=1}^N \{a^i\}_{i=1}^N, \mathcal{P}, R, \gamma)$. Each agent only has access to a partial observation of the global state and selects actions individually. Specifically, $\mathcal{N} = \{1, \ldots, N\}$ denotes the set of agents. $s \in S$ is the global state, $o^i$ is the observation of agent $i$ derived from $s$, and $a^i$ is the action of agent $i$. Notably, each observation $o^i$ can be decomposed into entity-wise features $\{e^1, e^2, \ldots, e^M\}$, where $M$ is the total number of entities (including both allies and enemies). The $\gamma \in [0, 1)$ is the discount factor. The transition function is $\mathcal{P} : s \times \boldsymbol{a} \times s \to [0, 1]$, where the joint action is defined as $\boldsymbol{a} = \{a^i\}_{i=1}^N$. The action-observation history is $\tau_t^i = (o_0^i, a_0^i, ..., o_t^i)$, which serves as input to the policy $\pi^i(a_t^i | \tau_t^i) \to [0, 1]$. At each timestep, the environment provides a reward $r_t$ via the reward function $\mathcal{R} : s \times \boldsymbol{a} \to \mathbb{R}$.

### 3.2 Multi-Task MARL: Policy Transfer via Offline Data

Although MARL algorithms have achieved substantial progress, they still require training from scratch for each task, which is time-consuming and interaction-heavy. Nevertheless, policies trained on one task often exhibit transferable behavioral patterns applicable to similar tasks. For example, coordination policies learned in the SMAC *3m* task can be reused in the *5m* setting. To measure how well such knowledge transfers, we adopt multi-task MARL as a principled benchmark. We denote $\mathcal{T}_{\text{src}} = \{T_{\text{src}}^n\}_{n=1}^{N_{\text{src}}}$ and $\mathcal{T}_{\text{tgt}} = \{T_{\text{tgt}}^n\}_{n=1}^{N_{\text{tgt}}}$ as the sets of source and unseen target tasks, respectively, where $N_{\text{src}}$ and $N_{\text{tgt}}$ represent the task numbers. Source tasks $\{T_{\text{src}}^n\}_{n=1}^{N_{\text{src}}}$ are associated with an offline dataset $\boldsymbol{\mathcal{D}}_{\text{src}} = \{\mathcal{D}_{\text{src}}^n\}_{n=1}^{N_{\text{src}}}$ collected by a pre–trained policy. A trajectory from $\mathcal{D}_{\text{src}}^n$ is defined as $(s_0^n, \boldsymbol{a_0^n}, r_0^n, \ldots, s_H^n)$, where $H$ is the length of trajectory, $s_t^n$ and $\boldsymbol{a_t^n}$ are the joint state and action

at time $t$, and $r_t^n = R^n(s_t^n, \boldsymbol{a}_t^n)$ is the reward. Under the multi-task generalization setting, our objective is to learn a general multi-agent policy $\pi$ that maximizes the expected discounted return across all tasks, shown in Eq. 1. After training, $\pi$ is evaluated on $\mathcal{T}_{\text{tgt}}$ without further fine–tuning.

$$\max_{\pi} \; \mathbb{E}_{T^n \sim \{\mathcal{T}_{\text{src}}, \mathcal{T}_{\text{tgt}}\}} \left[ \sum_{t=0}^{H} \mathbb{E}_{\boldsymbol{a}_t^n \sim \pi} \left[ \gamma^t \, R^n(s_t^n, \boldsymbol{a}_t^n) \right] \right] \tag{1}$$

### 3.3 Decision Transformer

The Decision Transformer (DT) [3] leverages the Transformer architecture to formulate the Markov decision process as a conditional sequence modeling problem, which attains high performance under offline training. Given the offline trajectories $\mathcal{D}$ consisting of states $s_t$, actions $a_t$, and rewards $r_t$, DT models the joint distribution over actions as: $a_t = f(\hat{R}_{t-L:t}, s_{t-L:t}, a_{t-L:t-1})$. $\hat{R}_t = \sum_{t'=t}^{H} r_{t'}$ denotes the target return-to-go, $L$ is the content length, and $f$ is the DT model which is trained to maximize the likelihood of the observed action sequences conditioned on past states, actions, and returns, shown in Figure 2. This formulation allows DT to learn directly from offline trajectories.

$$\max_{\pi} \mathbb{E}_{(\hat{R}_{\leq t}, s_{\leq t}, a_{< t}) \sim \mathcal{D}} \left[ \sum_{t=1}^{H} \log f(a_t \mid \hat{R}_{t-L:t}, s_{t-L:t}, a_{t-L:t-1}) \right] \tag{2}$$

## 4 Method

In this section, we introduce our novel **Bi**-Level **K**nowledge **T**ransfer (BiKT) for Multi-Task Multi-Agent Reinforcement Learning method. We argue that effective multi-task generalization in MARL necessitates knowledge transfer at two distinct levels: (1) The individual level, which involves learning and reusing agent-specific behaviors, referred to as individual skills $z$; and (2) The team level, which aims to transfer coordinated behavioral patterns that emerge from combinations of individual skills, referred to as cooperative tactics $c$. BiKT enables each agent to first generate a team-level tactic, and then infer an individual skill conditioned on it. Our objective is to learn both individual skill and team tactic embeddings from source tasks, utilize them to guide the decision making, and then effectively reuse them in unseen tasks.

Specifically, the BiKT framework for multi-task MARL is structured into three stages: (1) Extracting the individual skill embeddings for each agent that encapsulate transferable behaviors applicable across source tasks. (2) Building a cooperative tactic set, which captures coordination patterns of individual skills and supports knowledge transfer at the team level. (3) Developing a decision-making model that integrates both individual skills and team tactics to guide the policy execution. The following sections describe each stage in detail.

### 4.1 Knowledge Transfer through Individual Skills

In multi-agent scenarios, the abstraction of agent $i$'s policy at timestep $t$ is usually used to denote individual skill $z_t^i$. We aim to fulfill the skill learning process with two components: First, each agent can extract its individual skill embeddings from the state $s$ and joint actions $\boldsymbol{a}$, which captures common and reusable action patterns in source tasks. Second, when acquiring $z_t^i$, agent $i$ can reconstruct actions by its history $\tau_t^i$ to take decentralized execution. To achieve them, we model the individual skill learning process as the Individual Skill Encoder: $p_{\text{skill}}(\cdot|s, \boldsymbol{a}, i) \to z^i \in \mathbb{R}^{N_s}$, where $N_s$ is the dimension of skill embedding, and Action Decoder: $q_{\text{act}}(\cdot|\tau_t^i, z_t^i) \to \hat{a}^i$, illustrated in Figure 2. For skill training, we adopt the Variational Auto-Encoder (VAE) [15] framework. Our objective is to maximize the log-likelihood of reconstructing actions while regularizing the latent skill distribution, given by Equation 3, where $\delta_t^i = (s_t, \tau_t^i, \boldsymbol{a}_t), i \in N, t \in H, \tilde{p}(z_t^i)$ is the uniform prior over skills, $D_{KL}$ is the KL divergence function, and $\phi_1$ is the parameters of $p_{\text{skill}}$ and $q_{\text{act}}$.

$$\mathcal{L}_{\text{skill}}(\phi_1) = \mathbb{E}_{\delta_t^i \sim \mathcal{D}_{\text{src}}} \left[ \mathbb{E}_{z_t^i \sim p_{\text{skill}}(\cdot|s_t, \boldsymbol{a}_t, i)} \left[ \log q_{\text{act}}(\cdot|\tau_t^i, z_t^i) \right] - D_{KL} \left[ p_{\text{skill}}(\cdot|s_t, \boldsymbol{a}_t, i)||\tilde{p}(z_t^i) \right] \right], \tag{3}$$

Individual skills learning offers a task-agnostic abstraction over actions, effectively mitigating the challenge of action semantic misalignment across tasks. Besides, they serve as a commonly used knowledge and can be transferred across tasks. Our focus then shifts to leveraging cooperative patterns among skills to facilitate effective policy transfer at the team level.

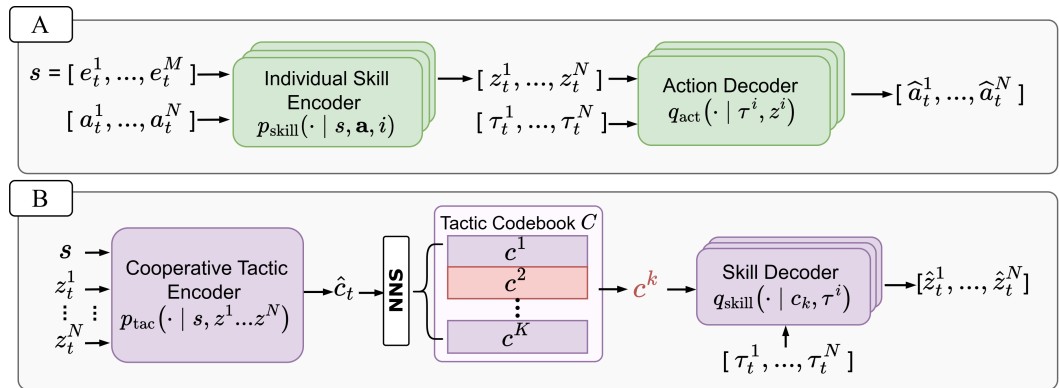

Figure 2: (A) The Individual Skill learning process, where $e_t^i$ stands for the feature of entity $i$. (B) illustrates the Cooperative Tactic codebook learning process. NNS stands for the **N**earest **N**eighbor **S**earch: Search the nearest neighbor from the codebook $\mathcal{C}$, i.e., $\arg\min_{k \in K} \|\hat{c}_t - c^k\|_2^2$. The network details and feature processing details about encoders and decoders are shown in the Appendix C.

## 4.2 Knowledge Transfer through Cooperative Tactics

While a single-agent can often generalize across tasks through reusable individual skills, multi-task MARL poses an additional challenge: Agents need not only to transfer individual skills but also to learn to coordinate them to achieve effective team-level behaviors across diverse tasks. Intuitively, the coordination of skills can be represented as a skill combination, structured as $(z^1 \times z^2 \times \cdots \times z^N)$. However, due to variations of the number of agents across tasks, the resulting skill combinations may differ even when the underlying skill combinations are the same. As shown in Figure 1, the "All focus fire to an enemy" tactic in *5m_vs_6m* and *7m_vs_8m* corresponds to different skill combinations, yet they semantically represent the same team tactic. To handle such variation, we propose a mapping process from a single tactic to multiple task-specific skill combinations, enabling the reuse of the same tactic across diverse task settings with varying agent numbers and environmental dynamics.

Specifically, we propose to map the skill combinations into a compact tactic set, denoted by *tactic codebook* $\mathcal{C} = \{c^k\}_{k=1}^K$, where the $c^k \in \mathbb{R}^{N_c}$ represents a reused cooperative tactic embedding, $N_c$ is the embedding length, and $K$ is the tactic number of $\mathcal{C}$. Each tactic $c^k$ is intended to represent a class of similar skill combinations across tasks and each agent can infer its individual skill based on the given $c^k$. To this end, we construct a Cooperative Tactic Encoder $p_{\text{tac}}(\cdot|s, \{z^i\}_{i=1}^N) \to \hat{c}_t \in \mathbb{R}^{N_s}$, a Skill Decoder $q_{\text{skill}}(\cdot|\tau_i, c^k) \to \hat{z}_t$, a the tactic codebook $\mathcal{C} = \{c^k\}_{k=1}^K$, shown in Figure 2. The Tactic Encoder maps the global state $s_t$ and joint individual skills $[z_t^1, \ldots, z_t^N]$ into a tactic representation $\hat{z}_t$, which is then discretized via the nearest neighbor search in the tactic codebook $\mathcal{C}$. The Skill Decoder then reconstructs each agent's skill from its local history $\tau^i$ and the tactic $c^k$.

To train the encoder, decoder, and tactic codebook, we adopt the Vector Quantized Variational Auto-Encoder (VQ-VAE) [34] framework to optimize two objectives: the reconstruction error of individual skills and the codebook commitment loss. The reconstruction loss minimizes the distance between the individual skills embedding $z^i$ and $\hat{z}^i$; The commitment loss encourages $p_{tac}$ to output close to the selected $c^k$, minimizing the distance between $\hat{c}_t$ and $c^k$. To promote diversity among tactics, we also impose a regularization term that penalizes similarity between tactics in $\mathcal{C}$. Our complete training objective is given by Equation 4, where $\zeta_t^i = (s_t, \tau_t^i, z_t^1, ..., z_t^N), i \in N, t \in H$, $\phi_2$ is the parameter set of $p_{\text{tac}}, q_{\text{skill}}$, and $\phi_c$ is the parameters of $\mathcal{C}$. $\text{sg}[\cdot]$ is the stop-gradient operator, $\beta_1, \beta_2$ balances the codebook commitment loss and distance penalty, $\epsilon$ is a small constant for numerical stability.

$$\mathcal{L}_{\text{tactic}}(\phi_2, \phi_c) = \mathbb{E}_{\zeta_t^i \sim \mathcal{D}_{\text{src}}}\left[ \left\|z_t^i - q_{\text{skill}}(\hat{c}_t + \text{sg}(c^k - \hat{c}_t), \tau^i)\right\|_2^2 + \left\|\text{sg}(\hat{c}_t) - c^k\right\|_2^2 \right.$$
$$\left. + \beta_1 \left\|\hat{c}_t - \text{sg}(c^k)\right\|_2^2 + \beta_2 \sum_{k_1 \neq k_2} \frac{1}{||c^{k_1} - c^{k_2}||_2^2 + \epsilon} \right] \qquad (4)$$

Through this design, each learned tactic represents a meaningful and reusable coordination pattern, and the resulting tactic codebook captures a diverse range of team behaviors across source tasks. As

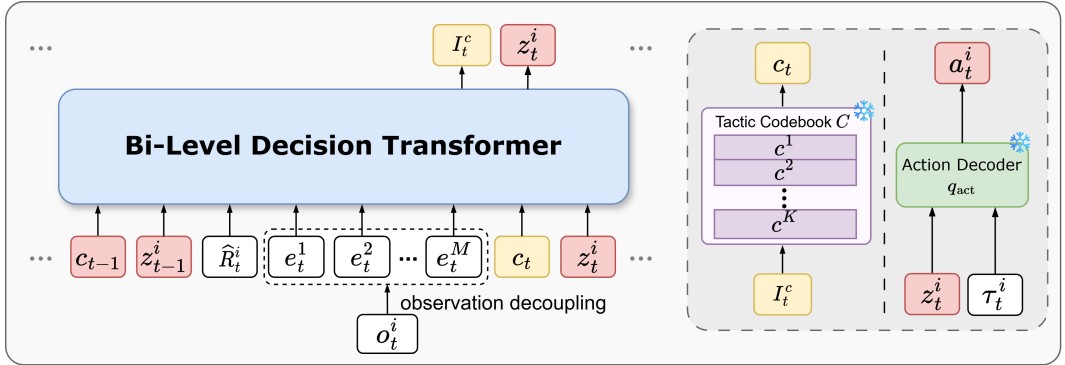

Figure 3: Bi-Level Decision Transformer (BDT) of agent $i$. At each timestep $t$, it receives a structured input sequence, including the index of the tactic codebook: $I_{<t}^c$, the previous skill $z_{<t}^t$, the predicted return-to-go $\hat{R}_t = \sum_{t=1}^{T} r_t$, and the agent's decoupled entity-wise observation $[e^1, e^2, \ldots, e^M]$. The BDT first predicts the index of the tactic codebook $I_t^c$, which is then used to retrieve the corresponding tactic $c_t$. Conditioned on the tactic $c_t$, the BDT generates its individual skill $z_t^i$, which is decoded into the action $a_t^i$ via the Action Decoder $q_{act}$.

a result, it enables flexible generalization to a broader set of unseen tasks. Moreover, these tactics act as stable team-level guidance, assisting agents in selecting appropriate individual skills across varying scenarios.

## 4.3 Decision Making with Tactics and Skills

After acquiring the individual skills and the tactic codebook, the remaining challenge lies in how to leverage them to guide policy transfer in unseen tasks. To this end, we propose our *Bi-Level Decision Transformer* (BDT) as the policy $\pi_\theta^i$ for agent $i$, which models an autoregressive distribution over the tactic $c_t \in \mathcal{C}$ and individual skills $z_t^i$: $\pi_\theta^i(c_t^i \mid \hat{R}_{\leq t}^i, o_{\leq t}^i, c_{<t}^i, z_{<t}^i)$ and $\pi_\theta^i(z_t^i \mid \hat{R}_{\leq t}^i, o_{\leq t}^i, c_{\leq t}^i, z_{<t}^i)$, detailed in Figure 3. To accommodate varying observation dimensions across tasks, raw observations are decomposed into structured entity representations to ensure a unified input format. Each agent learns to first generate a cooperative tactic, and then let them utilize it to take their individual skills.

The policy of each agent $\pi_\theta^i$ is trained in a supervised manner. We employ the pre-trained individual skill encoder $p_{skill}$ and the cooperative tactic encoder $p_{tac}$ to generate $z^i$ and $c^i$ as supervision signals. In practice, to promote tactic-level consistency, we train agent $i$ at timestep $t$ to predict the index of the shared tactic codebook $I_t^c$ rather than directly learning the tactic embeddings. The output dimension of $I_t^i$ corresponds to the number of tactics in $\mathcal{C}$. The training objective is defined in Equation 5, where $\Omega_t^i = (\hat{R}_{\leq t}^i, \tau_{<t}^i, c_{<t}^i, z_{<t}^i)$ and $\alpha$ is the hyperparameter to balance the learning speed for tactic and individual skill.

$$\mathcal{L}_{\text{policy}}(\theta) = \mathbb{E}_{\Omega_t^i \sim \mathcal{D}_{\text{src}}} \left[ -\alpha \log \pi_\theta^i(I_t^i \mid \hat{R}_{\leq t}^i, o_{\leq t}^i, c_{<t}^i, z_{<t}^i) - \log \pi_\theta^i(z_t^i \mid \hat{R}_{\leq t}^i, o_{\leq t}^i, c_{\leq t}^i, z_{<t}^i) \right] \quad (5)$$

It is worth noting that the design of DT-based policies for MARL has been previously explored, like MADT [23]. However, it is limited to single-task settings, as DT tends to learn a fixed policy pattern that often corresponds to a specific team combination. In contrast, our BDT design can utilize the team tactic as the team guidance to promote diverse team tactics, which extends the DT-based policy in MARL from a single task to multiple tasks.

**Overall Training Summary.** We first train the individual skill using $\mathcal{L}_{\text{skill}}(\phi_1)$ to acquire skill embeddings and the action decoder $q_{\text{act}}$. Next, we learn to construct the tactic codebook by minimizing $\mathcal{L}_{\text{tactic}}(\phi_2, \phi_c)$. At last, we leverage the pre-trained skill and tactic codebook to train the BDT by minimizing $\mathcal{L}_{\text{policy}}(\theta)$. The pseudocode is detailed in the Appendix 1.

**Decentralized Execution.** For each agent $i$, we initialize its policy BDT $\pi_\theta^i$ with a high return-to-go and input its history $\tau^i$ to predict the cooperative tactic codebook index $I_t^c$. Then, the corresponding tactic is retrieved from the codebook $\mathcal{C}$ and used to input into BDT to generate the individual skill. Finally, the individual skill is decoded into action $a_t^i$.

# 5 Experiment

We exploit different environments to conduct a large number of experiments, including StarCraft II Micromanagement (SMAC) [31] and Multi-Agent Particle Environment (MPE) [22]. We conduct five random seeds for each algorithm and evaluate them with 32 environments.

## 5.1 Experiment Setup

**StarCraft II Micromanagement** The StarCraft Multi-Agent Challenge (SMAC) [31] is a popular MARL benchmark which provides a standard platform for evaluating multi-task learning and policy transfer capabilities in MARL. Following the experimental protocol proposed by [41], we leverage the task set and corresponding offline datasets they released: *Marine-Hard*, *Marine-Easy*, and *Stalker-Zealot*. Each task set consists of distinct source tasks and unseen tasks. Specific details for each task set are presented in the Tables 3, 4, and 5. To evaluate generalization, we construct four groups in each task set: Expert, Medium, Medium-Expert, and Medium-Replay, representing different policy levels used for offline data collection, as detailed in Appendix B.1. Each group consists of multiple tasks that share the same unit type but differ in the number of units.

**Cooperative Navigation** The Cooperative Navigation (CN) task [22] is a widely adopted benchmark for evaluating MARL algorithms. In CN, agents should collaboratively occupy some target landmarks. The environment consists of $N$ agents and some landmarks in a two-dimensional continuous space. Agents must coordinate their physical actions to reach the landmarks. The objective is for each agent to cover a distinct landmark while avoiding collisions. To assess the generalization

Table 1: The performance of different methods in Task set *Marine-Hard*. To simplify notation, asymmetric tasks are abbreviated (e.g., "5m6m" represents "5m_vs_6m").

| Tasks | Expert | | | | Medium | | | |
|---|---|---|---|---|---|---|---|---|
| | UPDeT | ODIS | HiSSD | **BiKT** | UPDeT | ODIS | HiSSD | **BiKT** |
| | | | | *Source Tasks* | | | | |
| 3m | 82.8±16.0 | 98.4±2.7 | 99.5±0.3 | **99.8±0.2** | 51.2±3.4 | 85.9±10.5 | 62.7±5.7 | **86.1±9.6** |
| 5m6m | 17.2±28.0 | 53.9±5.1 | 78.5±4.5 | **80.3±4.5** | 6.3±4.9 | 22.7±7.1 | 26.4±3.8 | **36.9±4.5** |
| 9m10m | 3.1±5.4 | 80.4±8.7 | 95.5±2.7 | **99.4±0.4** | 28.5±10.2 | **78.1±3.8** | 73.9±2.3 | 72.5±5.5 |
| | | | | *Unseen Tasks* | | | | |
| 4m | 33.0±27.1 | 95.3±3.5 | 99.2±1.2 | **99.3±0.1** | 14.1±5.2 | 61.7±17.7 | 77.3±10.2 | **91.3±6.2** |
| 5m | 33.6±40.2 | 89.1±10.0 | 99.2±1.2 | **99.9±0.1** | 67.2±21.3 | 85.9±11.8 | 88.4±8.4 | **94.4±5.4** |
| 10m | 54.7±44.4 | 93.8±2.2 | 98.4±0.8 | **99.4±0.4** | 32.9±11.3 | 61.3±11.3 | **98.0±0.3** | 96.3±2.5 |
| 12m | 17.2±28.0 | 58.6±11.8 | 75.5±19.7 | **99.0±0.2** | 3.2±3.8 | 35.9±8.1 | 86.4±6.0 | **92.5±4.0** |
| 7m8m | 0.0±0.0 | 25.0±15.1 | 35.3±9.8 | **68.0±9.9** | 0.0±0.0 | **28.1±22.0** | 14.2±10.1 | 4.3±2.5 |
| 8m9m | 0.0±0.0 | 19.6±6.0 | 47.0±6.2 | **50.0±6.2** | 2.3±2.6 | **47.0±2.7** | 15.3±2.8 | 17.0±1.9 |
| 10m11m | 0.0±0.0 | 42.4±7.2 | 86.3±14.6 | **90.6±1.1** | 4.0±3.4 | 29.7±15.4 | **43.6±4.6** | 25.6±4.6 |
| 10m12m | 0.0±0.0 | 1.6±1.6 | 14.5±9.1 | **14.6±3.5** | 0.0±0.0 | 1.6±1.6 | 0.6±0.5 | **2.2±1.3** |
| 13m15m | 0.0±0.0 | 2.3±2.6 | 1.3±2.5 | **4.2±2.1** | 0.0±0.0 | 1.6±1.6 | 1.4±2.4 | **2.5±1.3** |
| | Medium-Expert | | | | Medium-Replay | | | |
| | | | | *Source Tasks* | | | | |
| 3m | 85.2±17.9 | 73.6±22.0 | 86.6±3.7 | **99.4±1.3** | 41.4±20.1 | **83.6±14.0** | 78.8±5.3 | **78.7±6.3** |
| 5m6m | 1.6±1.6 | 9.4±2.2 | 41.9±9.7 | **49.4±6.0** | 0.8±1.4 | 16.6±4.7 | 25.3±10.3 | **28.1±3.1** |
| 9m10m | 24.3±18.7 | 31.3±14.5 | **83.6±6.9** | 58.3±2.4 | 0.8±1.4 | 34.4±8.0 | 45.8±3.9 | **47.6±7.0** |
| | | | | *Unseen Tasks* | | | | |
| 4m | 43.9±39.0 | 82.8±13.5 | 91.1±6.1 | **98.1±0.2** | 35.9±12.6 | 55.6±14.5 | 77.3±1.9 | **94.4±2.8** |
| 5m | 33.6±40.2 | 82.8±17.7 | 98.3±1.8 | **98.8±0.1** | 61.7±20.3 | 96.1±4.1 | 88.1±13.4 | **97.9±0.2** |
| 10m | 32.8±38.1 | 82.8±16.8 | 96.4±2.1 | **96.5±1.2** | 11.0±7.8 | 84.4±15.1 | 94.7±2.6 | **96.7±1.5** |
| 12m | 9.4±8.6 | 81.3±20.6 | 88.4±11.8 | **95.6±0.3** | 2.3±2.6 | 84.4±6.6 | 90.3±3.6 | **92.7±2.6** |
| 7m8m | 2.3±4.1 | 15.6±4.4 | 30.5±10.4 | **40.6±4.2** | 1.6±2.7 | 9.4±2.2 | **21.7±4.7** | 19.3±6.4 |
| 8m9m | 10.2±4.6 | 9.5±8.6 | 10.9±4.7 | **22.4±3.6** | 11.5±3.9 | 0.8±1.4 | **11.7±8.7** | 10.5±4.0 |
| 10m11m | 11.8±8.1 | 33.6±8.9 | **54.7±6.8** | 42.3±3.2 | 0.8±1.4 | 35.9±5.2 | 42.5±4.4 | **43.8±6.6** |
| 10m12m | 0.0±0.0 | 1.6±1.6 | **2.5±1.0** | 2.3±1.3 | 0.0±0.0 | 2.4±1.4 | 0.5±0.3 | **0.6±0.2** |

Table 2: The performance of different methods in Task set *CN*.

| | Expert | | | | Medium | | | |
| | Source Tasks | | Unseen Tasks | | Source Tasks | | Unseen Tasks | |
| | CN-2 | CN-4 | CN-3 | CN-5 | CN-2 | CN-4 | CN-3 | CN-5 |
|---|---|---|---|---|---|---|---|---|
| UPDeT | 90.6±6.8 | 15.6±9.2 | 47.9±10.3 | 2.1±2.9 | 35.4±12.1 | 4.2±2.9 | 14.6±3.9 | 0.0±0.0 |
| ODIS | **100.0±0.0** | 46.2±13.6 | 85.6±7.6 | 20.0±7.8 | 65.0±5.4 | **28.7±6.7** | 43.8±5.2 | **8.1±2.5** |
| HiSSD | 96.4±2.8 | 49.2±7.2 | 89.8±5.2 | 25,3±2.9 | 59.3±5.2 | 24.2±3.9 | 44.9±2.9 | 5.1±2.9 |
| **BiKT** | **100.0±0.0** | **62.5±5.0** | **93.8±3.2** | **28.2±6.3** | **68.1±6.8** | 28.3±4.3 | **46.8±5.2** | 7.1±4.7 |

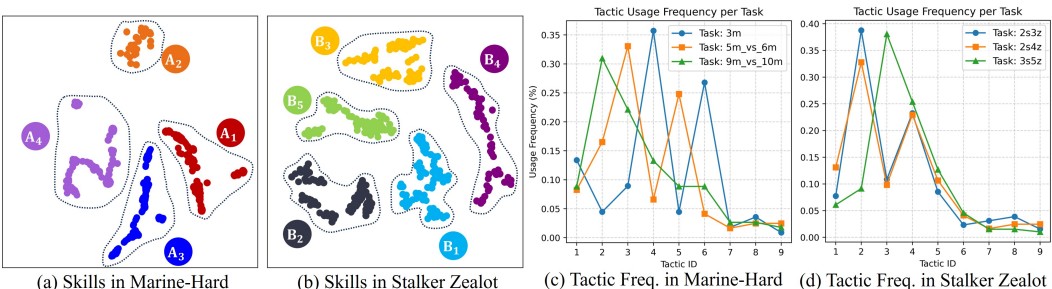

(a) Skills in Marine-Hard    (b) Skills in Stalker Zealot    (c) Tactic Freq. in Marine-Hard    (d) Tactic Freq. in Stalker Zealot

Figure 4: (a) and (b) represent discovered individual skill embeddings from source tasks. Each color represents a different individual skill type, which can be reused in different tasks. (c) and (d) represent the frequency of tactic usage per task from source tasks. Each tactic ID corresponds to a distinct team tactic. Representative skill and tactic examples are shown in Figure 5.

performance, the task set is denoted by *CN* and each task is denoted by *CN*-N, where N denotes the number of agents. Our training uses offline datasets provided by [41].

## 5.2 Performance Comparisons

**Baselines** We compare BiKT against representative baselines. 1. UPDeT, which extends the transformer-based UPDeT architecture by introducing a transformer-based Q-mixing network to enable effective multi-task policy learning. 2.ODIS [41], which leverages coordination skills extracted from offline multi-task data and learns to differentiate agent behaviors. 3. HiSSD [20], which employs a hierarchical framework that jointly learns shared and task-specific skills across multiple tasks. All methods are trained from source tasks and evaluated directly on unseen tasks.

**Overall, our method outperforms baselines.** The results for task set *Marine-Hard* and *CN* are shown in Table 1, 2. For paper limits, the results of *Stalker-Zealot* and *Marine-Easy* are detailed in the Appendix 9, 10. In task set *Marine Hard*, our method achieves superior performance compared to baselines, especially on the Expert and Medium-Expert datasets. In contrast, UPDeT simply relies on network design to enable multi-task learning, without explicitly considering policy transfer. This limits its ability to generalize beyond simple tasks like 3m. While ODIS performs well on source tasks, its success is limited to similar tasks(e.g., 3m, 4m, 5m,...). It fails to transfer performance from 5m_vs_6m to more distinct unseen tasks such as 7m_vs_8m and 8m_vs_9m, especially under low-quality offline datasets. HiSSD learns to distinguish common and specific skills across, which implicitly serve as task-specific guidance information. In contrast, our method explicitly learns cooperative tactics from skill combinations, resulting in superior performance.

**Ablation Study** We detail them in the Appendix D.3 due to the paper limits.

## 5.3 Strengths of BiKT

**Our method can discover common individual skills.** Figure 4 shows the t-SNE visualization of the learned individual skill embeddings for the source tasks of *Marine-Hard* and *Stalker-Zealot*. It does not differentiate the skills from different tasks, which we include in Appendix D.2. Trajectories are collected using the trained BiKT policy. Figure 5 explains several semantics of these skills, and more explanations for each skills are provided in the Appendix D.4. From the visualizations, we can conclude that the skills exhibit a clear clustering trend, which reflects a shared and reusable skill set Besides, the embeddings of the same skill type may vary across tasks for different action adaptation.

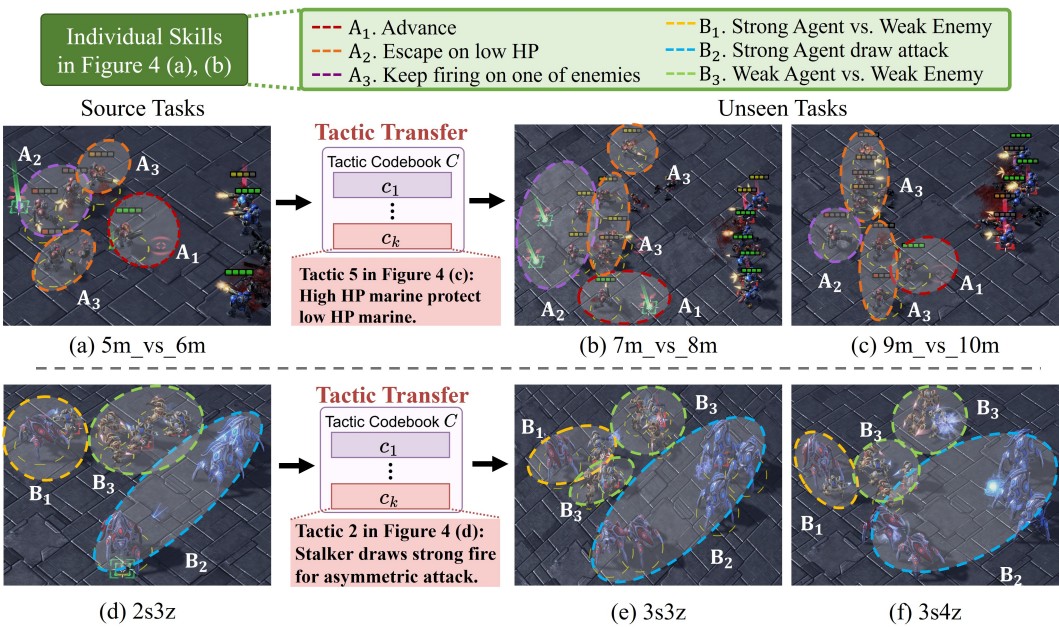

Figure 5: Examples of tactic transfer in different task sets. In case (a), there exists unbalanced health point (HP) among the agents, the high HP agent takes the skill $A_1$ to advance, the low HP agents take skill $A_3$ to escape and other agents take skill $A_2$ to keep firing. It corresponds to the tactic that a high HP agent advances to draw firepower and protect the team, which is effectively transferred to tasks *7m_vs_8m* and *9m_vs_10m*. In case (d), one stalker takes the skill $B_2$ to draw attack from stalker enemies, the other stalker takes the skill B1 to attack the weaker zealot enemy, and the rest zealots take the skill $B_3$ to attack enemies. Considering that the stalkers are more powerful than zealots, this forms the key tactic 2 for winning, which is been learned to transfer to task 3s3z and 3s4z.

**Our method can discover diverse cooperative tactics.** We collect 32 trajectories of trained policies in different tasks and track the frequency of discovered team tactics. Figure 4 presents the usage distribution of each tactic ID under the *Marine-Hard* and *Stalker-Zealot* task sets. We conclude that certain team tactics are consistently utilized across all tasks. However, some tactics vary significantly in frequency depending on the task, often serving as key factors for success. *team focus fire* of tactic 2 is preferred in scenarios with numerical disadvantage (e.g., *5m_vs_6m*), helping to quickly eliminate threats, while *local team fire* of tactic 4 suits simpler tasks like *3m*. These results indicate that our method selects tactics adaptively, depending on task-specific requirements.

**Explicable example: Our method performs policy transfer by bi-level knowledge.** Figure 5 shows two examples illustrating how our method enables policy transfer by reusing learned tactics. In both cases, the policy leverages specific tactics learned from source tasks and successfully generalizes them to unseen tasks, facilitating coordinated team behavior. Notably, the team cooperative information in these two examples cannot be transferred by skills from the individual level. This demonstrates that agents leverage the same tactic to infer similar skill combination across tasks.

# 6 Conclusion

In this paper, we aim to tackle the problem of multi-task MARL generalization via offline data. Though existing works utilize skill-based methods to achieve knowledge transfer, we argue that transferring skills alone is not efficient. We think the efficient knowledge transfer in multi-task MARL includes two levels: the individual level that transfers skills and the team level that transfers cooperative tactics. To fulfill it, we introduce our BiKT method, which first discovers latent skill representations from offline trajectories using a VAE, then constructs a discrete tactic codebook via VQ-VAE, and finally learns a policy with a Bi-Level Decision Transformer that select tactic id and executes skills sequentially. Experimental results demonstrate that BiKT significantly improves the zero-shot generalization performance of mutli-task MARL.

# 7 Acknowledgements

This work was supported in part by the National Key R&D Program of China (No. 2025ZD0122000), NSFC 62273347, the Key Research and Development Program of Jiangsu Province (BE2023016).

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

# A PSEUDOCODE of our method

The pseudocode of our method is detailed in 1.

---
**Algorithm 1** BiKT for Multi-Task MARL
---
1: **Inputs:**
2: The offline dataset $\boldsymbol{\mathcal{D}}_{\text{src}} = \{\mathcal{D}_{\text{src}}^n\}_{n=1}^{N_{\text{src}}}$, Individual Skill Encoder $p_{\text{skill}}$, Action Decoder $q_{\text{act}}$, cooperative tactic encoder $p_{\text{tac}}$, skill decoder $q_{\text{skill}}$, the tactic codebook embeddings $\mathcal{C} = \{c_k\}_{k=1}^K$, the skill-based decision transformer $\{\pi_i\}_{i=1}^N$, the number of source tasks $N_{\text{src}}$, the number of agents $N$, the tactic codebook number $K$, the content length of bi-level decision transformer $L$, learning rates $l_1, l_2, l_3$
3: **Training:**
4: **for** each timestep $t$ in $1..H$ **do**
5:     Sample $\delta_t^i = (s_t, \tau_t^i, \boldsymbol{a}_t) \sim \boldsymbol{\mathcal{D}}_{\text{src}}$       # Individual Skill Learning
6:     Use $\delta_t^i$ to compute $\mathcal{L}_{\text{skill}}(\phi_1)$ in Eq. 3.
7:     Calculate gradients to update $p_{\text{skill}}$ and $q_{\text{act}}$, with learning rate $l_1$
8: **end for**
9: **for** each timestep $t$ in $1..H$ **do**
10:     Sample $\zeta_t^i = (s_t, \tau_t^i, z_t^1, ..., z_t^N) \sim \boldsymbol{\mathcal{D}}_{\text{src}}$     # Cooperative Tactic Codebook Learning
11:     Use $\zeta_t^i$ to compute $\mathcal{L}_{\text{tactic}}(\phi_2)$ in Eq. 4.
12:     Calculate gradients to update $p_{\text{tac}}$ and $q_{\text{skill}}$, with learning rate $l_2$
13: **end for**
14: **for** each timestep $t$ in $1..H$ **do**
15:     Sample $\Omega_t^i = (\tau_{\leq t}^i, c_{<t}^i, z_{<t}^i, \hat{R}_t^i) \sim \boldsymbol{\mathcal{D}}_{\text{src}}$     # Bi-level Decision Transformer Learning
16:     Use $\Omega_t^i$ to compute $\mathcal{L}_{\text{policy}}(\theta)$ in Eq. 5.
17:     Calculate gradients to update $p_{\text{skill}}$ and $q_{\text{act}}$, with learning rate $l_3$
18: **end for**
19: **Execution:**
20: **for** each timestep $t$ in source task $\mathcal{T}_{\text{src}}^n$ **do**
21:     Given return-to-go $\{\hat{R}_t^i\}_{i=1}^N$
22:     $\{c_t^i\}_{i=1}^N \leftarrow \pi_\theta^i(c_t^i \mid \hat{R}_{\leq t}^i, o_{\leq t}^i, c_{<t}^i, z_{<t}^i)$     # Select team tactic
23:     $\{z_t^i\}_{i=1}^N \leftarrow \pi_\theta^i(z_t^i \mid \hat{R}_{\leq t}^i, o_{\leq t}^i, c_{\leq t}^i, z_{<t}^i)$     # Take Individual skills
24:     $\{a_t^i\}_{i=1}^N \leftarrow q_{\text{act}}(\cdot|, \tau_t^i, ..., \tau_t^N, z_t^1, ..., z_t^N)$     # Take Individual actions
25: **end for**
---

# B Experiment Setting Details

## B.1 SMAC

**Environment Overview**    SMAC is derived from the real-time strategy game StarCraft II, focusing on micromanagement. Unlike typical StarCraft II games that involve both macromanagement (strategic planning) and micromanagement (fine control of units), SMAC is structured to emphasize decentralized control by requiring each unit to be managed by an independent agent based solely on local, limited observations. This setup necessitates multi-agents learning sophisticated cooperative behaviors under the challenge of partial observability. SMAC consists of diverse micro scenarios designed to assess how well agents coordinate to solve complex tasks. Each scenario involves two opposing armies with variations in initial positioning, unit types, and terrain features.

**Observations, Actions and Team Goal.**    At each timestep, agents gain local observations within their field of view, providing information such as distance, health, shields, and unit type of visible units, as well as terrain features. During centralized training, the global state includes comprehensive data on all units, including energy levels and attack cooldowns. Agents have a discrete action set including movement, attacks, healing by Medivacs with certain constraints ensuring decentralization. The shooting range of units is limited compared to their sight range, necessitating strategic maneuvering. The allied units are controlled by agents trained to maximize the win rate against enemy units governed by the game's AI using scripted strategies.

**Multi-Task Settings.**    To assess multi-task generalization, we follow the ODIS setting with three task sets: *Marine-Hard*, *Marine-Easy*, and *Stalker-Hard*. Each task set consists of distinct training

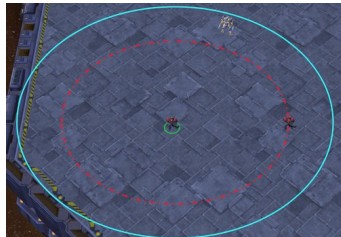 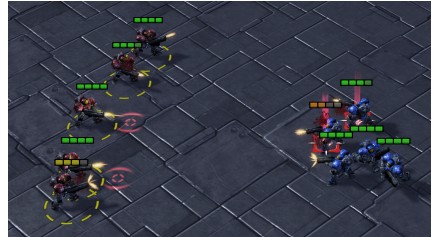 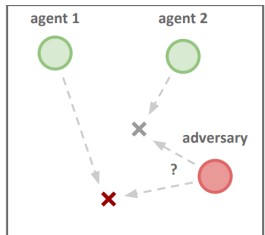

(a) SMAC Agent Local Observation      (b) The Map 5m of SMAC      (c) Cooperatiave Navigation

Table 3: Descriptions of tasks in the Marine-Hard task set.

| Task type | Task | Ally units | Enemy units | Properties |
|---|---|---|---|---|
| Source tasks | 3m | 3 Marines | 3 Marines | homogeneous & symmetric |
|  | 5m_vs_6m | 5 Marines | 6 Marines | homogeneous & asymmetric |
|  | 9m_vs_10m | 9 Marines | 10 Marines | homogeneous & asymmetric |
| Unseen tasks | 4m | 4 Marines | 4 Marines | homogeneous & symmetric |
|  | 5m | 5 Marines | 5 Marines | homogeneous & symmetric |
|  | 10m | 10 Marines | 10 Marines | homogeneous & symmetric |
|  | 12m | 12 Marines | 12 Marines | homogeneous & symmetric |
|  | 7m_vs_8m | 7 Marines | 8 Marines | homogeneous & asymmetric |
|  | 8m_vs_9m | 8 Marines | 9 Marines | homogeneous & asymmetric |
|  | 10m_vs_11m | 10 Marines | 11 Marines | homogeneous & asymmetric |
|  | 10m_vs_12m | 10 Marines | 12 Marines | homogeneous & asymmetric |
|  | 13m_vs_15m | 13 Marines | 15 Marines | homogeneous & asymmetric |

source tasks and testing scenarios. Specific details for each task set are presented in the Tables 3, 4 and 5. The *Marine-Hard* and *Marine-Easy* sets comprise different marine battle scenarios where the learned multi-agent strategy must guide allied marines against enemy marines controlled by the game's AI, matching or exceeding in number. The *Stalker-Zealot* set consists of several challenges involving equal numbers of stalkers and zealots on either side.

**Offline Dataset.** In our experiments, we use the same dataset collected by ODIS for the fair comparison. Within each task set, the offline data is collected by pre-trained QMIX policies with different levels of performance: *Expert*, *Medium*, *Medium-Expert* and *Medium-Replay*. The *Expert* policy is trained with 2,000,000 environment steps. The *Medium* policy is trained until it achieves approximately a 50The *Medium-Expert* dataset is a mixture of trajectories from both the *Expert* and *Medium* policies. The *Medium-Replay* dataset is obtained from the replay buffer of the *Medium* policy, which contains a larger proportion of lower-quality trajectories. Table 6 summarizes the full settings of the offline datasets.

### B.2 Cooperaitve Navigation

**Environment Overview** To further evaluate our method, we consider a task set based on the Cooperative Navigation (CN) scenario, a representative cooperative task from the Multi-Agent Particle Environment (MPE). The environment consists of $N$ agents and $L$ landmarks situated in a two-dimensional continuous space with discrete time steps. Agents must coordinate their physical actions to reach the $L$ landmarks. Each agent observes the relative positions of other agents and landmarks, and the team receives a shared reward based on the proximity of any agent to each landmark—that is, the goal is for all landmarks to be 'covered' by the team. Agents occupy physical space and are penalized for collisions with one another, encouraging coordinated but non-overlapping behaviors. In this setting, agents must infer which landmark to cover and navigate there while avoiding others. The agents can execute discrete actions of moving towards four directions and a "none" operation.

Table 4: Descriptions of tasks in the Marine-Easy task set.

| Task type | Task | Ally units | Enemy units | Properties |
|---|---|---|---|---|
| Source tasks | 3m | 3 Marines | 3 Marines | homogeneous & symmetric |
| | 5m | 5 Marines | 5 Marines | homogeneous & symmetric |
| | 10m | 10 Marines | 10 Marines | homogeneous & symmetric |
| Unseen tasks | 4m | 4 Marines | 4 Marines | homogeneous & symmetric |
| | 6m | 6 Marines | 6 Marines | homogeneous & symmetric |
| | 7m | 7 Marines | 7 Marines | homogeneous & symmetric |
| | 8m | 8 Marines | 8 Marines | homogeneous & symmetric |
| | 9m | 9 Marines | 9 Marines | homogeneous & symmetric |
| | 11m | 11 Marines | 11 Marines | homogeneous & symmetric |
| | 12m | 12 Marines | 12 Marines | homogeneous & symmetric |

Table 5: Descriptions of tasks in the Stalker-Zealot task set.

| Task type | Task | Ally units | Enemy units | Properties |
|---|---|---|---|---|
| Source tasks | 2s3z | 2 Stalkers, 3 Zealots | 2 Stalkers, 3 Zealots | heterogeneous & symmetric |
| | 2s4z | 2 Stalkers, 4 Zealots | 2 Stalkers, 4 Zealots | heterogeneous & symmetric |
| | 3s5z | 3 Stalkers, 5 Zealots | 3 Stalkers, 5 Zealots | heterogeneous & symmetric |
| Unseen tasks | 1s3z | 1 Stalkers, 3 Zealots | 1 Stalkers, 3 Zealots | heterogeneous & symmetric |
| | 1s4z | 1 Stalkers, 4 Zealots | 1 Stalkers, 4 Zealots | heterogeneous & symmetric |
| | 1s5z | 1 Stalkers, 5 Zealots | 1 Stalkers, 5 Zealots | heterogeneous & symmetric |
| | 2s5z | 2 Stalkers, 5 Zealots | 2 Stalkers, 5 Zealots | heterogeneous & symmetric |
| | 3s3z | 3 Stalkers, 3 Zealots | 3 Stalkers, 3 Zealots | heterogeneous & symmetric |
| | 3s4z | 3 Stalkers, 4 Zealots | 3 Stalkers, 4 Zealots | heterogeneous & symmetric |
| | 4s3z | 4 Stalkers, 3 Zealots | 4 Stalkers, 3 Zealots | heterogeneous & symmetric |
| | 4s4z | 4 Stalkers, 4 Zealots | 4 Stalkers, 4 Zealots | heterogeneous & symmetric |
| | 4s5z | 4 Stalkers, 5 Zealots | 4 Stalkers, 5 Zealots | heterogeneous & symmetric |

**Multi-Task settings.** The task set of CN consists of different numbers of agents. Specifically, CN-$n$ denotes a CN map containing $n$ agents. Offline datasets are collected using the QMIX algorithm. Detailed dataset settings are summarized in Table 7.

### B.3 Computing Resources

For computing resources, we utilize the *Intel(R) Xeon(R) Gold 5220* CPU and *NVIDIA TITAN RTX* GPU in the experiments. Each experiment in per task set lasts on average for 8 hours.

## C  Implementation Details

In this section, we will provide the model structure, the hyperparameters, and other training details of ODIS. We present each part of BiKT in the following sections.

Table 6: Properties of offline datasets in SMAC with different qualities.

| Tasks | Quality | Trajectories | Average Return | Average Win Rate |
|---|---|---|---|---|
| 3m | expert | 2000 | 19.8929 | 0.9910 |
| | medium | 2000 | 13.9869 | 0.5402 |
| | medium-expert | 4000 | 16.9399 | 0.7656 |
| | medium-replay | 3603 | N/A | N/A |
| 5m | expert | 2000 | 19.9380 | 0.9937 |
| | medium | 2000 | 17.3288 | 0.7411 |
| | medium-expert | 4000 | 18.6334 | 0.8674 |
| | medium-replay | 711 | N/A | N/A |
| 10m | expert | 2000 | 19.9438 | 0.9922 |
| | medium | 2000 | 16.6297 | 0.5413 |
| | medium-expert | 4000 | 18.2595 | 0.7626 |
| | medium-replay | 571 | N/A | N/A |
| 5m_vs_6m | expert | 2000 | 17.3424 | 0.7185 |
| | medium | 2000 | 12.6408 | 0.2751 |
| | medium-expert | 4000 | 14.9916 | 0.4968 |
| | medium-replay | 32607 | N/A | N/A |
| 9m_vs_10m | expert | 2000 | 19.6140 | 0.9431 |
| | medium | 2000 | 15.5049 | 0.4146 |
| | medium-expert | 4000 | 17.5594 | 0.6789 |
| | medium-replay | 13731 | N/A | N/A |
| 2s3z | expert | 2000 | 19.7655 | 0.9602 |
| | medium | 2000 | 16.6279 | 0.4465 |
| | medium-expert | 4000 | 18.1967 | 0.7034 |
| | medium-replay | 4505 | N/A | N/A |
| 2s4z | expert | 2000 | 19.7402 | 0.9509 |
| | medium | 2000 | 16.8735 | 0.4965 |
| | medium-expert | 4000 | 18.3069 | 0.7237 |
| | medium-replay | 6172 | N/A | N/A |
| 3s5z | expert | 2000 | 19.7850 | 0.9518 |
| | medium | 2000 | 16.3126 | 0.3114 |
| | medium-expert | 4000 | 18.0488 | 0.6316 |
| | medium-replay | 11528 | N/A | N/A |

Table 7: Properties of offline datasets in Cooperative Navigation with different qualities.

| Tasks | Quality | Trajectories | Average Return | Average Win Rate |
|---|---|---|---|---|
| CN-2 | expert | 2000 | 1.0000 | 1.0000 |
| | medium | 2000 | 0.6152 | 0.6152 |
| CN-4 | expert | 2000 | 0.7173 | 0.7173 |
| | medium | 2000 | 0.4273 | 0.4273 |

## C.1 Multi-Head Attention

We utilize Multi-Head Attention (MHA) to represent the embeddings of skills and tactics. This mechanism enables the model to jointly attend to different representation subspaces, making it effective for modeling contextual dependencies. Given query, key, and value matrices $Q \in \mathbb{R}^{T \times d}$, $K \in \mathbb{R}^{S \times d}$, and $V \in \mathbb{R}^{S \times d}$, the scaled dot-product attention is computed as in Equation 6. Multi-head attention applies this operation across $h$ heads, where each head has its own projection matrices $W_i^Q$, $W_i^K$, and $W_i^V$. The result of each head is shown in Equation 7, and the final output is formed

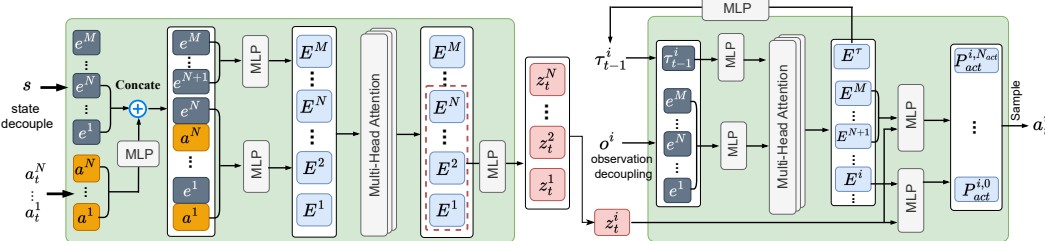

Figure 6: The detailed model structure of our individual skill learning.

by concatenating all heads and applying a linear projection, as shown in Equation 8.

$$\text{Attention}(Q, K, V) = \text{softmax}\left(\frac{QK^\top}{\sqrt{d_k}}\right)V \tag{6}$$

$$\text{head}_i = \text{Attention}(QW_i^Q, KW_i^K, VW_i^V) \tag{7}$$

$$\text{MHA}(Q, K, V) = \text{Concat}(\text{head}_1, \ldots, \text{head}_h)W^O \tag{8}$$

## C.2  Details of Individual Skill Learning

The detailed architecture for individual skill learning is illustrated in Figure 6. During the skill encoding phase, each agent employs the shared encoder $p_{\text{skill}}(\cdot \mid s, \boldsymbol{a}, i)$ to compute its latent skill embedding $z^i \in \mathbb{R}^{N_s}$. Specifically, the encoder takes as input the global state $s$, the joint action $\boldsymbol{a}$, and the agent index $i$.

To construct the encoder input, we first concatenate the actions and entity features of the $N$ allied agents. These $N$ concatenated representations, along with the remaining $M - N$ entity features (e.g., enemies or neutral units), are mapped into a unified set of $M$ embeddings. These embeddings are then passed through the Multi-Head Attention module of the Transformer. Finally, we take the $N$ attention outputs corresponding to the allied agents and feed them into an MLP to generate the individual skill embeddings.

To reconstruct actions, each agent is expected to infer its action based on its trajectory history and individual skill embedding. To achieve this, we first extract entity features from the observation $o^i$, and process the history $\tau_t^i$ and entity features through separate multilayer perceptrons (MLPs). The resulting representations are then fed into a MHA module to capture relevant contextual dependencies. The MHA output is subsequently concatenated with the individual skill embedding $z_t^i$ and passed through another set of MLPs to produce the action logits. These logits define the action distribution $P_{\text{act}}$ over $N_{\text{act}}$ discrete action dimensions. Finally, the action is sampled from this distribution for execution.

## C.3  Details of Cooperative Tactic Learning

The detailed architecture for the cooperative tactic codebook is illustrated in Figure 7. We construct the *Cooperative Tactic Encoder* $p_{\text{tac}}(\cdot \mid s, \{z^i\}_{i=1}^N) \to \hat{c}_t \in \mathbb{R}^{N_s}$, the *Skill Decoder* $q_{\text{skill}}(\cdot \mid \tau^i, c_k) \to \hat{z}_t^i$, and the tactic codebook $\mathcal{C} = \{c_k\}_{k=1}^K$. The tactic embeddings in codebook $\mathcal{C}$ is randomly initialized.

The Tactic Encoder maps the global state $s_t$ and the individual skill embeddings $[z_t^1, \ldots, z_t^N]$ into a continuous tactic representation $\hat{c}_t$. Specifically, each individual skill $z_t^i$ is concatenated with the corresponding ally's entity features, forming $N$ enriched ally representations. These, along with the remaining entity features, are separately embedded into vectors and then fed into the MHA module. The outputs are averaged to produce the final tactic embedding. This tactic embedding is then discretized by searching the nearest neighbor in the tactic codebook $\mathcal{C}$, resulting in the selected tactic $c_k \in \mathcal{C}$.

To decode the individual skills, the entity features extracted from observations and agent $i$'s trajectory history $\tau^i$ are fed into the skill decoder $q_{\text{skill}}$. These inputs are first embedded separately using MLPs and then passed through the MHA module. The output embedding corresponding to agent $i$ and the individual skill representation $z_t$ are passed through an MLP to generate the final decoded skill.

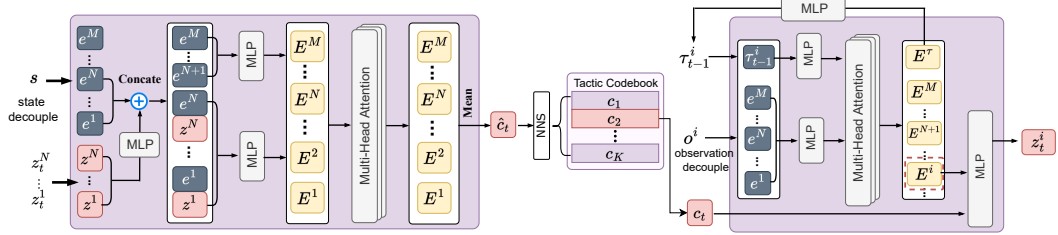

Figure 7: The detailed model structure of our cooperative tactic codebook.

Subsequently, the Skill Decoder reconstructs each agent's latent skill $\hat{z}_t^i$ from its local trajectory history $\tau^i$ and the selected tactic $c_k$. This process enables the integration of global cooperative strategies with decentralized agent behavior during execution.

During the tactic encoding phase, each agent employs the encoder $p_{\text{tac}}(\cdot \mid s, \boldsymbol{a}, i)$ to compute its latent skill embedding $z^i \in \mathbb{R}^{N_s}$. Specifically, the encoder takes as input the global state $s$, the joint action $\boldsymbol{a}$, and the agent index $i$.

**Stop-Gradient Operator.** The stop-gradient operator, denoted as $\text{sg}[\cdot]$, is used to block gradients during backpropagation while preserving values during the forward pass. Formally, for a variable $x$, the stop-gradient operation behaves as:

$$\text{Forward:} \quad \text{sg}[x] = x, \qquad \text{Backward:} \quad \frac{\partial\,\text{sg}[x]}{\partial x} = 0. \tag{9}$$

That is, in the forward pass, $\text{sg}[x]$ evaluates to the same value as $x$, but during the backward pass, no gradients are propagated through $\text{sg}[x]$.

## C.4 Hyper-parameters of BiKT

The hyperparameters of our method are detailed in Table 8

Table 8: Hyperparameters of our method.

| Hyperparameter | Value |
|---|---|
| Individual skill dimension $N_s$ | 4 |
| Tactic embedding $N_c$ | 64 |
| The number of tactics in $\mathcal{C}$: $K$ | 16 |
| Hidden layer dimension of BDT | 64 |
| The multi-head number of BDT | 2 |
| The content length of BDT | 10 |
| Optimizer | Adam |
| Training steps for $L_{\text{skill}}$ | 15000 |
| Training steps for $L_{\text{tactic}}$ | 8000 |
| Training steps for $L_{\text{policy}}$ | 30000 |
| Batch size | 32 |
| learning rate $l_1$ | 0.0004 |
| learning rate $l_2$ | 0.0001 |
| learning rate $l_3$ | 0.0002 |
| $\beta_1$ | 1 |
| $\beta_2$ | 0.01 |
| $\alpha$ | 0.05 |

Table 9: The performance of different methods in Task set *Stalker Zealot*.

| Tasks | Expert | | | | Medium | | | |
|---|---|---|---|---|---|---|---|---|
| | UPDeT-m | ODIS | Hi-SSD | **BiKT** | UPDeT-m | ODIS | Hi-SSD | **BiKT** |
| *Source Tasks* | | | | | | | | |
| 2s3z | 50.0±33.4 | 97.7±2.6 | 95.2±1.0 | **97.9±2.3** | 35.0±23.0 | 49.2±8.4 | 32.3±11.7 | **51.6±3.3** |
| 2s4z | 23.4±26.6 | 60.9±6.8 | 79.8±6.0 | **93.2±5.1** | 18.8±10.3 | **32.8±12.2** | 17.0±2.2 | 25.0±7.4 |
| 3s5z | 17.2±19.8 | 87.5±9.6 | 92.8±5.0 | **93.0±4.5** | 25.6±24.2 | 28.9±6.8 | 24.4±7.9 | **29.8±2.3** |
| *Unseen Tasks* | | | | | | | | |
| 1s3z | 1.6±1.6 | 76.6±3.5 | **81.6±15.2** | 77.0±4.2 | 3.8±5.0 | 41.4±18.8 | **44.2±9.9** | 32.4±0.4 |
| 1s4z | 26.6±19.3 | 17.2±10.5 | 42.0±26.1 | **52.6±15.1** | 2.5±3.6 | **50.7±7.5** | 18.1±11.0 | 22.6±0.5 |
| 1s5z | **29.7±26.4** | 2.5±2.3 | 16.7±12.3 | 8.7±4.3 | 5.0±4.2 | 14.1±8.4 | 2.5±2.2 | **18.8±3.9** |
| 2s5z | 23.4±22.2 | 27.3±6.0 | **79.7±2.2** | 75.3±3.2 | 16.9±14.1 | **32.0±4.6** | 11.3±3.7 | 24.2±6.5 |
| 3s3z | 20.3±10.9 | 89.1±5.2 | 88.0±4.5 | **98.4±1.6** | 24.4±28.6 | 23.4±9.2 | 21.9±10.7 | **34.4±2.2** |
| 3s4z | 12.5±19.9 | 96.9±2.2 | 88.1±9.0 | **97.7±2.3** | 28.8±31.6 | 50.8±15.5 | 17.2±4.5 | **54.8±0.5** |
| 4s3z | 6.2±4.4 | 64.1±13.0 | 88.6±4.1 | **96.9±2.3** | 11.2±18.0 | 13.3±7.5 | **31.9±23.2** | 18.7±0.1 |
| 4s4z | 7.8±13.5 | **79.7±10.9** | 73.4±5.2 | 75.5±5.3 | 1.2±1.5 | 12.5±7.0 | 13.2±6.5 | **16.7±1.4** |
| 4s5z | 5.5±7.8 | **86.7±12.6** | 65.6±3.7 | 44.5±6.8 | 5.6±8.5 | 7.0±4.1 | 4.5±1.3 | **8.3±1.4** |
| 4s6z | 4.7±6.4 | **88.3±8.4** | 68.4±4.9 | 68.2±6.9 | 1.9±2.5 | 1.6±1.6 | 0.9±0.9 | **2.5±2.5** |
| | **Medium-Expert** | | | | **Medium-Replay** | | | |
| *Source Tasks* | | | | | | | | |
| 2s3z | 57.5±27.1 | 58.6±15.5 | 68.1±8.1 | **81.3±7.4** | 14.4±13.2 | 15.6±18.2 | 9.0±1.5 | **30.2±8.4** |
| 2s4z | **53.1±24.6** | 41.4±7.8 | 41.9±10.2 | **73.8±7.8** | 12.5±9.7 | 7.8±5.2 | 6.0±1.2 | **30.8±7.1** |
| 3s5z | 35.0±23.5 | 41.4±18.5 | 57.8±10.7 | **59.1±8.7** | 20.0±16.6 | 18.8±3.1 | 17.5±2.0 | **19.3±7.1** |
| *Unseen Tasks* | | | | | | | | |
| 1s3z | 4.4±8.8 | 72.7±12.2 | 73.0±10.2 | **75.9±9.1** | 0.0±0.0 | 21.1±20.4 | **36.3±7.1** | 30.9±9.1 |
| 1s4z | 11.9±9.8 | **44.5±20.3** | 32.3±30.5 | 37.9±5.9 | 7.5±10.0 | 6.2±7.7 | 24.8±9.1 | **26.3±7.2** |
| 1s5z | 3.8±4.6 | **42.2±31.4** | 9.4±9.5 | 14.4±19.4 | 11.9±9.6 | 7.8±6.4 | 4.4±2.2 | **12.5±4.7** |
| 2s5z | 37.5±22.5 | **43.0±10.7** | 25.6±7.8 | 19.0±5.2 | **20.0±16.8** | 14.1±8.1 | 16.5±2.8 | 17.2±8.4 |
| 3s3z | 33.8±15.0 | 50.0±13.3 | 56.6±25.6 | **57.9±8.2** | 17.5±12.3 | 25.0±20.1 | 9.6±3.3 | **27.6±4.5** |
| 3s4z | 43.1±20.7 | 52.3±9.5 | 71.7±9.7 | **75.6±13.3** | 15.6±11.2 | 19.5±16.6 | **22.5±10.6** | 19.4±11.1 |
| 4s3z | 23.8±21.0 | 17.2±7.2 | **60.5±15.1** | 28.8±9.4 | 11.2±15.0 | 8.6±14.9 | **11.0±10.4** | 10.4±5.1 |
| 4s4z | 10.6±13.8 | 20.3±6.8 | 37.3±9.4 | **39.9±4.9** | 5.6±9.8 | 4.7±8.1 | **9.4±1.8** | 8.3±2.9 |
| 4s5z | 11.9±16.1 | 21.9±2.2 | 17.0±4.1 | **24.3±5.3** | **10.6±19.7** | 0.8±1.4 | 0.8±0.8 | 4.4±3.5 |
| 4s6z | 5.0±8.5 | 18.0±5.1 | **19.7±5.9** | 14.8±3.2 | **6.9±13.8** | 2.3±4.1 | 2.3±4.1 | 3.5±2.9 |

Table 10: The performance of different methods in Task set *Marine Easy*

| Tasks | Expert | | | | Medium | | | |
|---|---|---|---|---|---|---|---|---|
| | UPDeT-m | ODIS | Hi-SSD | **BiKT** | UPDeT-m | ODIS | Hi-SSD | **BiKT** |
| *Source Tasks* | | | | | | | | |
| 3m | 83.6±12.6 | 97.7±2.6 | **99.5±8.1** | 99.4±1.3 | 60.2±29.9 | 57.8±9.2 | 74.7±14.6 | **87.2±4.7** |
| 5m | 74.8±22.9 | 95.3±5.2 | **99.9±0.0** | **99.9±0.0** | 67.8±5.9 | **82.8±5.2** | 81.6±10.8 | 74.2±5.9 |
| 10m | 83.6±19.2 | 88.3±20.3 | 95.2±8.4 | **99.9±0.0** | 48.8±7.9 | 71.9±6.6 | **84.8±8.6** | 82.1±7.2 |
| *Unseen Tasks* | | | | | | | | |
| 4m | 53.0±32.3 | 90.6±7.0 | 94.4±2.9 | **96.9±1.5** | 41.7±17.4 | 63.3±16.1 | 74.5±15.5 | **75.5±8.6** |
| 6m | 37.9±8.6 | 79.7±17.5 | 99.7±0.3 | **99.7±0.1** | 75.8±22.7 | **89.8±17.6** | 88.0±10.0 | 83.0±4.7 |
| 7m | 44.2±13.2 | 72.7±16.9 | 99.1±0.7 | **99.7±0.1** | 65.2±25.2 | 96.1±1.4 | **97.3±2.3** | 89.9±0.0 |
| 8m | 51.7±26.2 | 80.9±14.4 | **99.8±0.3** | 99.1±0.1 | 88.4±13.7 | 97.7±2.6 | 93.8±5.2 | **98.9±1.2** |
| 9m | 76.3±13.4 | 99.2±1.4 | **99.9±0.0** | **99.9±0.0** | 64.8±35.6 | 87.5±2.2 | 75.2±15.5 | **88.9±11.8** |
| 11m | 53.6±22.4 | 83.6±12.4 | 99.2±0.8 | **99.3±1.0** | 23.4±11.8 | 64.7±3.1 | 62.0±21.8 | **68.2±4.7** |
| 12m | 44.3±22.8 | 70.3±30.2 | **99.7±1.1** | 99.6±1.0 | 13.5±11.7 | 41.4±6.0 | 55.5±25.7 | **49.7±13.4** |
| | **Medium-Expert** | | | | **Medium-Replay** | | | |
| *Source Tasks* | | | | | | | | |
| 3m | 48.4±36.8 | 89.8±9.7 | 90.9±5.9 | **91.3±4.8** | 29.7±10.0 | 79.7±4.7 | **87.7±2.9** | 78.8±3.2 |
| 5m | 64.1±17.9 | 83.7±16.0 | 79.4±6.9 | **85.3±5.9** | 6.2±10.8 | 3.1±5.4 | 87.5±2.9 | **88.5±1.6** |
| 10m | 68.8±23.8 | **93.8±4.4** | 60.2±21.1 | 83.6±3.1 | 0.0±0.0 | 0.0±0.0 | 84.2±4.9 | **85.2±2.5** |
| *Unseen Tasks* | | | | | | | | |
| 4m | 43.7±25.0 | 57.8±18.8 | 70.9±9.1 | **72.2±8.3** | 25.0±22.6 | 25.0±5.4 | 71.6±4.1 | **77.2±4.7** |
| 6m | 47.7±30.0 | 76.0±6.0 | 70.6±6.1 | **78.2±1.6** | 0.0±0.0 | 3.1±5.4 | **99.8±0.3** | 86.8±3.2 |
| 7m | 57.8±32.9 | 66.4±14.6 | 85.0±11.7 | **85.6±16.4** | 0.0±0.0 | 0.0±0.0 | **99.8±0.3** | 84.2±1.4 |
| 8m | 40.6±19.3 | 43.8±11.5 | **72.8±9.5** | 68.3±4.6 | 0.0±0.0 | 1.6±1.6 | **96.7±0.3** | 87.6±1.7 |
| 9m | 47.7±24.8 | 73.4±16.2 | **80.0±14.6** | 70.8±6.6 | 0.0±0.0 | 0.0±0.0 | **88.8±1.3** | 86.6±2.4 |
| 11m | **85.9±14.2** | 68.8±20.3 | 70.9±5.9 | 75.5±12.4 | 0.0±0.0 | 0.0±0.0 | 45.6±4.5 | **52.3±3.6** |
| 12m | 46.1±15.5 | 62.5±8.0 | **62.7±7.8** | 59.7±9.8 | 0.0±0.0 | 0.0±0.0 | 38.0±3.7 | **41.5±4.3** |

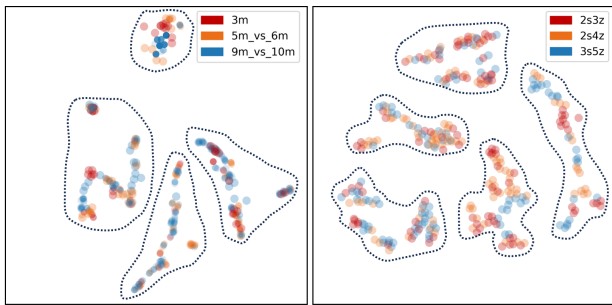

(a) Skills in Marine-Hard    (b) Skills in Stalker Zealot

Figure 8: The individual skill embeddings.

# D    Additional Results

## D.1    The performance comparison of other task sets.

The results for task set *Stalker Zealot* and *Marine Easy* are shown in Table 9 10. The *Stalker Zealot* requires different tactics in different tasks, which brings big challenge in policy transfer. The results show that our BiKT overall outperforms other baselines in both task sets. However, in *Marine Easy*, the tactics required for each task are similar, resulting that all methods can achieve high performance. It makes that Hi-SSD and BiKT can both obtain high performance in Expert setting.

## D.2    Visualization of individual skills from source tasks

We additionally show the individual skill embeddings from the task labels, in Figure 8.

## D.3    Ablation study

We conduct ablation experiments on task set *Marine Hard* to evaluate different variants of our method, and the results are shown in Table 11.

- MADT_w_OD: To evaluate the impact of tactic and skill learning, we remove them and let the decision transformer learns to take action directly, which naturally degrades into the MADT method with observation decoupling, denoted by MADT_w_OD. For fairness, we utilize the same hyperparameter in MADT_w_OD. For convenience we also provide the results of MADT.

- $L = 5$: We set the context length of skill-based Decision Transformer $\pi_i$ as 5.

- $\mathcal{C}_{K=32}$: During the team tactic learning process, we set the tactic number of codebook $K$ as 32.

- w/o $\mathcal{C}$: We overpass the learning process of team tactic and directly let the skill-based decision transformer to learn the individual skills. It is achieved by removing the $c_t^i$ embedding token in Figure 3. At this time, the SDT policy learns to directly output the individual skills and then takes its action.

- Con-Tac: Compared with continuous individual skill embeddings, we utilize a fixed number of tactics. For that we employ a VAE to learn team-level tactics and use *Continuous Tactic* (Con-Tac) embeddings instead of a discrete tactic set.

**Ablation study: The action based policy struggles to generalize to different tasks.**    The results of MADT_w_OD in Table 11 indicate that our proposed tactic and skill learning components play a crucial role in the overall performance. Using an action-based policy introduces the challenge that the agent must take different actions under similar observations across diverse tasks, which cannot be addressed effectively without additional guiding information. As a result, MADT performs well only in tasks with similar numbers of agents and comparable problem settings, such as *3m*, *4m*, and *5m*. Although the observation decoupling in MADT_w_OD leads to performance improvements, it is not the primary contributing factor to generalization.

**Ablation Study: Individual Skills Alone Cannot Transfer Diverse Team-Level Knowledge**   The results of *w/o C* in Table 11 show that without the guidance of team tactics, our method's performance drops. This is because the policy must execute different skills without any external team-level information. In this setting, agents tend to learn a fixed combination of skills that only adapts well to a limited set of tasks, such as *3m*, *4m*, and *5m*. However, this approach fails to capture diverse team tactics required for more complex tasks like *7m_vs_8m* and *8m_vs_9m*, leading to a noticeable decline in policy transfer performance .

**Ablation Study: Discrete Tactic Codebook Outperforms Continuous Tactic Embeddings**   The results for Con-Tac in Table 11 indicate that using continuous tactic embeddings can improve performance on some tasks. However, it still falls short of the results achieved with a discrete tactic codebook (BikT). We argue that each tactic should correspond to a clear, meaningful, and reusable coordination pattern. Different tasks often share common tactics, which provide stable team-level guidance and assist agents in selecting appropriate individual skills across varied scenarios. Continuous tactic embeddings tend to blur this clarity, thereby weakening the effectiveness of team strategies.

**Ablation Study: Impact of Content Length and Tactic Codebook Size**   The results with a content length $L = 5$ for the BDT model demonstrate that our method's success does not heavily depend on a complex Transformer architecture. Instead, the key factor is the way we incorporate bi-level knowledge transfer in multi-task MARL, which proves to be highly effective. Regarding the tactic codebook size, the results with $C = 32$ show that our tactic learning process converges to meaningful and useful tactic embeddings. This indicates that a moderately sized codebook is sufficient to achieve both efficiency and performance.

Table 11: The results of ablation study in task set *Marine Hard*

| | Source Tasks | | | Unseen Tasks | | |
|---|---|---|---|---|---|---|
| | 3m | 5m_vs_6m | 9m_vs_10m | 4m | 5m | 10m |
| MADT | 88.5±3.9 | 3.1±0.0 | 1.0±1.5 | 83.3±5.3 | 75.0±6.8 | 1.0±1.5 |
| MADT_w_OD | 90.2±2.8 | 10.2±3.5 | 16.2±6.8 | 88.4±2.3 | 83.2±2.7 | 12.7±4.2 |
| $L = 5$ | 100.0±0.0 | 78.9±3.5 | 98.4±1.5 | 99.3±0.9 | 99.9±0.8 | 97.2±1.5 |
| $C_{K=32}$ | 100.0±0.0 | 80.9±3.2 | 99.2±0.3 | 99.2±0.1 | 99.3±0.2 | 99.3±0.2 |
| w/o $C$ | 98.2±0.2 | 68.2±5.3 | 83.2±2.5 | 90.2±1.9 | 88.6±2.7 | 86.2±3.2 |
| Con-Tac | 99.2±0.4 | 64.3±6.7 | 80.3±4.4 | 92.2±2.4 | 90.6±4.6 | 88.3±4.2 |
| BiKT | **100.0±0.0** | **81.3±4.5** | **99.4±0.4** | **99.3±0.1** | **100.0±0.0** | **99.4±0.1** |
| | Unseen Tasks | | | | | |
| | 12m | 7m_vs_8m | 8m_vs_9m | 10m_vs_11m | 10m_vs_12m | 13m_vs_15m |
| MADT | 0.0±0.0 | 0.0±0.0 | 0.0±0.0 | 0.0±0.0 | 0.0±0.0 | 0.0±0.0 |
| MADT_w_OD | 8.2±3.2 | 8.4±3.9 | 0.0±0.0 | 0.0±0.0 | 0.0±0.0 | 0.0±0.0 |
| $L = 5$ | 98.7±0.9 | 64.8±8.2 | 48.2±8.3 | 90.2±2.3 | 12.0±1.8 | 3.2±1.5 |
| $C_{K=32}$ | 99.0±0.2 | 70.2±8.0 | 46.2±7.4 | 90.2±2.1 | 12.3±2.2 | 3.3±1.6 |
| w/o $C$ | 57.2±4.2 | 23.3±5.5 | 20.3±9.2 | 40.7±9.2 | 1.6±1.6 | 1.6±1.3 |
| Con-Tac | 60.5±3.8 | 18.2±7.4 | 17.8±5.2 | 38.7±8.3 | 0.9±0.5 | 0.5±0.4 |
| BiKT | **99.0±0.2** | **68.0±9.9** | **50.0±6.2** | **90.6±1.1** | **14.6±1.5** | **4.2±2.1** |

## D.4   Semantic of Individual skills and tactics

We provide more examples for our learned skills and tactics, as shown in Figure 9, 10 and  12.

# E   Discissions

**We adopt different strategies for learning tactic embeddings and skill embeddings, each with a distinct focus.**   For tactic embeddings, which serve as a shared team-level guiding signal for all agents, we represent them as fixed or discrete values. This design enables all agents to access the same team guidance, facilitating coordinated cooperation. Additionally, we employ VQ-VAE to learn

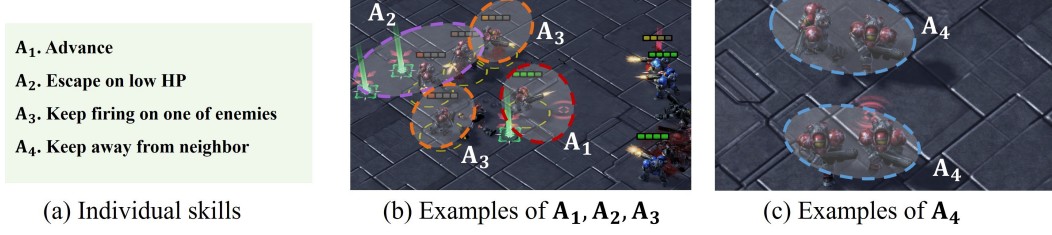

| | |
|---|---|
| **A₁.** Advance | |
| **A₂.** Escape on low HP | |
| **A₃.** Keep firing on one of enemies | |
| **A₄.** Keep away from neighbor | |

(a) Individual skills   (b) Examples of $A_1, A_2, A_3$   (c) Examples of $A_4$

Figure 9: The semantics of individual skills in *Marine-Hard*

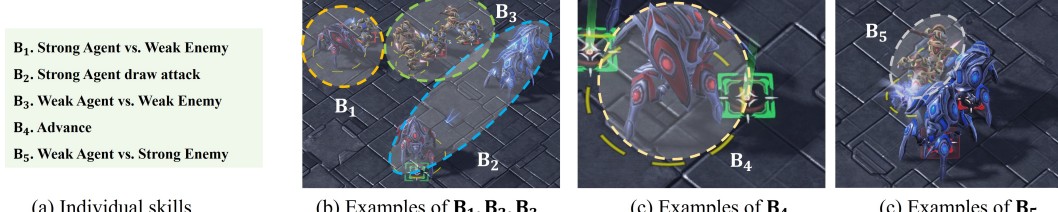

| | |
|---|---|
| **B₁.** Strong Agent vs. Weak Enemy | |
| **B₂.** Strong Agent draw attack | |
| **B₃.** Weak Agent vs. Weak Enemy | |
| **B₄.** Advance | |
| **B₅.** Weak Agent vs. Strong Enemy | |

(a) Individual skills   (b) Examples of $B_1, B_2, B_3$   (c) Examples of $B_4$   (c) Examples of $B_5$

Figure 10: The semantics of individual skills in *Stalker-Zealot*.

a compact set of tactic embeddings across multiple MARL tasks, allowing the discovery of a limited set of reusable team strategies. In contrast, using a standard VAE for tactic learning would result in a unique team guidance vector for each trajectory in every task, introducing higher randomness and instability when generalizing to unseen tasks. For skill embeddings, we use a VAE to preserve the diversity and distributional structure of the skill embedding space. The agents' action semantics vary across environments and agent numbers. Even when two agents in different scenarios exhibit similar behaviors or skills, we still expect their skill embeddings to differ. Using VAE allows for these embeddings to be mapped to actions with different physical meanings in different environment maps.

# F   Limitations

While our proposed method demonstrates strong generalization across the evaluated tasks, it remains an open question whether it can consistently maintain performance when scaled to highly diverse and large-scale task distributions. Exploring more expressive or adaptive embedding mechanisms could be a promising direction for future work.

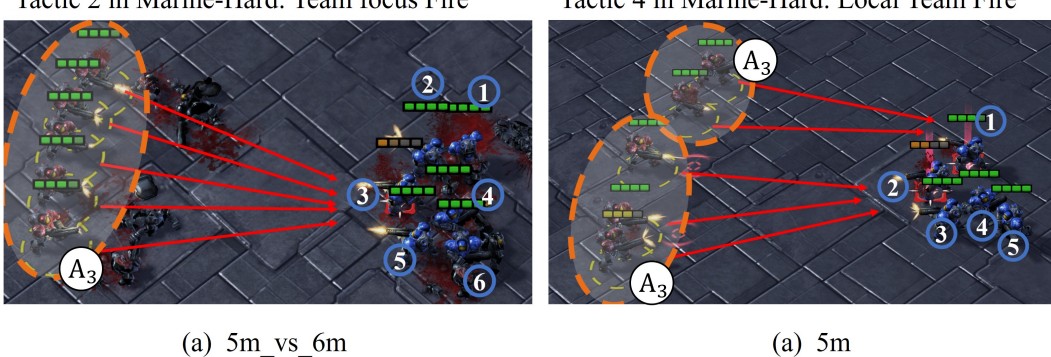

Tactic 2 in Marine-Hard: Team focus Fire

Tactic 4 in Marine-Hard: Local Team Fire

(a) 5m_vs_6m

(a) 5m

Figure 11: The semantics of some tactics. The tactic ids correspond to Figure 4, which are learned from the offline trajectories. In (a), agents are learned to take all fire to a single enemy target, which can quickly eliminate an enemy and make up for the disadvantage in agent numbers. This tactic is more aggressive, and the win rate is not stable. In (b), the agents are learned to attack their enemy targets locally, without considering the disadvantage in agent numbers. It provides a more stable win rate, but it falls in 5m_vs_6m.

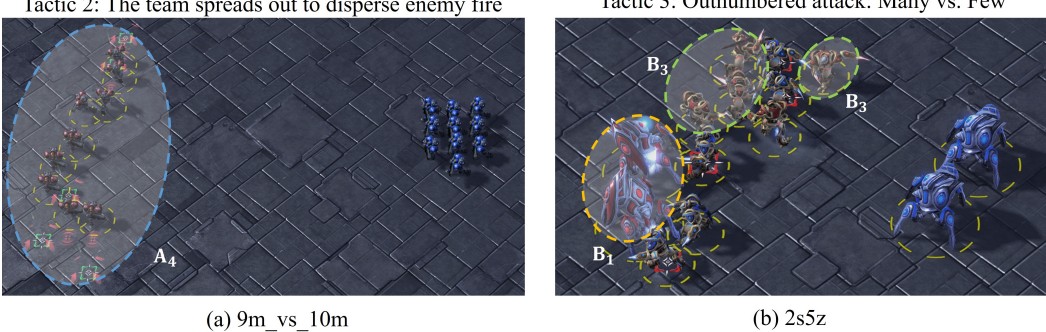

Tactic 2: The team spreads out to disperse enemy fire

Tactic 3: Outnumbered attack: Many vs. Few

(a) 9m_vs_10m

(b) 2s5z

Figure 12: The semantics of some tactics. The tactic ids correspond to Figure 4. In (a), agents are learned to take skill $A_4$ to keep away from neighbors, which forms the team tactic that the team spreads out to disperse enemy fire. In (b), the agents learn to take skills $B_1$ and $B_3$, allowing the stronger agent to attack a weaker enemy, while the weaker agent also targets a weak enemy. This tactic leverages numerical superiority to quickly eliminate weaker opponents.

