# OpenReview forum: "Bi-Level Knowledge Transfer for Multi-Task Multi-Agent Reinforcement Learning"
_NeurIPS.cc/2025/Conference — NeurIPS 2025 poster_

### Official Review · Reviewer_cecZ · 2025-06-23

**Clarity:** 4
**Significance:** 3
**Originality:** 3
**Rating:** 5
**Confidence:** 3

**Summary:**

This paper presents a novel framework for zero-shot generalization across multiple tasks in multi-agent reinforcement learning (MARL) using offline data. The core insight is that successful policy reuse in MARL should not only transfer individual agent skills but also team-level coordination patterns, referred to as cooperative tactics. To this end, the authors introduce a Bi-Level Knowledge Transfer (BLKT) algorithm that decouples and encodes these two forms of knowledge using VQ-VAE-based embeddings and a decision-transformer-style backbone.

I find the proposed approach conceptually sound and cleverly simple. While the building blocks are composed of established methods, the design is well-motivated, clearly articulated, and effectively implemented. The experiments demonstrate strong performance and offer qualitative insights into the role of coordination in multi-agent generalization.

**Questions:**

I do not have additional major questions for the authors beyond the clarification regarding return-to-go initialization. The paper is well-executed, methodologically sound, and clearly presented. I am confident in my recommendation to accept and believe the contribution will be of interest to the MARL and offline RL communities.

**Ethical Concerns:**

["NO or VERY MINOR ethics concerns only"]

**Limitations:**

yes

**Paper Formatting Concerns:**

No concerns

**Quality:**

3

**Strengths And Weaknesses:**

## Strengths

- **Clear motivation and novel framing**: The paper highlights a key insight in MARL generalization, namely, that coordinated team-level behaviors should be learned and transferred explicitly. This two-level abstraction (skills vs. tactics) is both intuitive and well-executed.
- **Strong engineering and thoughtful design**: Although the components used (e.g., VQ-VAE, nearest neighbor search, decision transformers) are not novel individually, their combination is technically solid and contributes to a well-functioning pipeline.
- **Clarity and structure**: The writing is excellent, concise, easy to follow, and logically structured. Figures (especially Figure 1) are helpful and informative.
- **Empirical performance**: The method shows strong empirical results, and the evaluation setup is well designed and clearly explained in Section 3.
- **Qualitative analysis**: I particularly appreciated the additional insights offered through Figure 4 (skill/tactic visualization) and the qualitative assessments in the results section and Table 2. These help substantiate the benefits of the proposed decomposition.

---

## Weaknesses

- **Lack of algorithmic novelty**: While the method is well-motivated and effective, it primarily integrates existing techniques without introducing fundamentally new algorithms. However, given the strength of the empirical results and the thoughtful architectural integration, this is not a significant drawback in this case.

- **Initialization details could be clarified**: In Section 3.2, for each agent \(i\), the BDT is initialized with a "high return-to-go." However, it’s unclear how this value is chosen. I guess this is environment-dependent and can be inferred from the offline dataset. I would appreciate if the authors could provide some additional details on this point as that would help clarify some assumptions.

---

> ### Author Rebuttal · Authors · 2025-07-29
>
> We sincerely appreciate your recognition of our methodology, experimental results, and overall presentation, as well as the insightful feedback you've provided. Below, we offer detailed clarifications in response to your comments, and we hope our revisions help resolve any concerns and enhance the strength of our work.
>
> > Weakness1:  While the method is well-motivated and effective, it primarily integrates existing techniques without introducing fundamentally new algorithms.
>
> We would like to clarify that our main contribution lies in the Bi-level Knowledge Transfer framework, which provides a novel perspective on policy transfer in multi-agent reinforcement learning (MARL).  Instead of treating policy transfer as a monolithic process, we explicitly decompose it into team-level tactic transfer and individual-level skill transfer. This design offers several advantages:
>
> -  **Bi-level Knowledge provides essential guidance for policy transfer of MARL.**  Policy transfer in  MARL fundamentally requires agents to execute appropriate team policies across a variety of unseen tasks. Given that all agents must share the same policy model while adapting to different tasks, a team-level guiding signal is needed to steer agents toward task-specific policy preferences. We define this as the learning of tactic-level knowledge. Meanwhile, from the perspective of each agent’s local policy, certain behaviors or decision patterns can be reused across tasks. We refer to this transferable component as individual skill knowledge. Together, these two types of knowledge form the important foundation for effective policy transfer in MARL.
>
> - **Structured Knowledge transfer mitigates learning challenges.**  Tactics focus on team-level strategy learning, while individual skills emphasize the abstraction of local actions.
> This separation enables the tactic codebook to capture the diverse patterns of skill composition across different tasks, summarizing the variability of team behaviors. Meanwhile, individual skill learning extracts reusable local action patterns across tasks, highlighting their generality.  The bi-level knowledge transfer structure alleviates the challenges of multi-task learning by introducing an explicit organization of knowledge.
>
> - **Flexible team-level generalization.**  Through BDT, our method selects appropriate tactics from the tactic codebook in different environments and enables agents to execute the corresponding skills, thereby enabling coordinated decision-making across different tasks.
>
> While our current implementation of BiKT is built upon existing modules, it is primarily intended to **validate the effectiveness of the bi-level knowledge transfer paradigm**. We believe that each module (e.g., skill encoder, tactic encoder) is modular and can be replaced with alternative techniques to suit different environments and multi-task settings. Therefore, the novelty of our work lies in proposing and validating a generalizable and modular framework for hierarchical knowledge transfer in MARL, which can inspire future extensions and adaptations.
>
> > Weakness 2/ Question 1: In Section 3.2, for each agent (i), the BDT is initialized with a "high return-to-go." However, it’s unclear how this value is chosen. I guess this is environment-dependent and can be inferred from the offline dataset. I would appreciate if the authors could provide some additional details on this point as that would help clarify some assumptions.
>
> Your suggestion to clarify the return-to-go (rtg) setting is very helpful.
>
> To clarify, the setting of return-to-go is not our main contribution, and we adopt the default return-to-go setting as the previous works: Decision Transformer [1], [2], [3]. During evaluation, we set the target return based on the desired performance — for example, using the maximum possible return to encourage expert-like behavior.
>
> Specifically, the rtg is environment-dependent.  In fact, in SMAC and MPE,  the maximum rtgs in different maps are the same, which are 20 and 1, respectively, corresponding to a win rate of 100%. So we uniformly set the maximum return across all maps, including source tasks and unseen tasks.
>
> In addition, we conducted an ablation study on the Marine-Expert task, with several alternative rtg configurations:
>
> * The higher return-to-go: 25, denoted by C1
> * The average return of all trajectories, denoted by C2 (19.8929 for 3m; 17.3425 for 5m6m; 19.6140 for 9m10m)
>
> The results are shown below.
>
> | Different rtg | source-task 3m  | source-task 5m6m | source-task 9m10m | unseen-task 4m | unseen-task 5m  | unseen-task 10m | unseen-task 12m | unseen-task 7m8m | unseen-task 8m9m | unseen-task 10m11m | unseen-task 10m12m | unseen-task 13m15m |
> | ------------- | --------------- | ---------------- | ----------------- | -------------- | --------------- | --------------- | --------------- | ---------------- | ---------------- | ------------------ | ------------------ | ------------------ |
> | C1            | $100.0 \pm 0.0$ | $80.4 \pm 5.6$   | $98.4 \pm 1.1$    | $99.4 \pm 0.3$ | $100.0 \pm 0.0$ | $100.0 \pm 0.0$ | $98.8 \pm 0.4$  | $67.5 \pm 12.8$  | $51.3 \pm 6.8$   | $89.8 \pm 0.1$     | $13.8 \pm 3.6$     | $4.4 \pm2.4$       |
> | C2            | $100.0 \pm 0.0$ | $81.8 \pm 4.6$   | $98.2 \pm 1.5$    | $98.8 \pm 0.2$ | $100.0 \pm 0.0$ | $99.4 \pm 0.1$  | $98.8 \pm 0.4$  | $71.3 \pm 12.9$  | $47.5 \pm 8.4$   | $90.4 \pm 4.0$     | $12.5 \pm 4.9$     | $3.1 \pm0.3$       |
> | BiKT          | $100.0 \pm 0.0$ | $81.3 \pm 4.5$   | $99.4 \pm 0.4$    | $99.3 \pm 0.1$ | $100.0 \pm 0.0$ | $99.4 \pm 0.1$  | $99.0 \pm 0.2$  | $68.0 \pm 9.9$   | $50.0 \pm 6.2$   | $90.6 \pm 1.1$     | $14.6 \pm 1.5$     | $4.2 \pm 2.1$      |
>
> **Conclusion**: The results show that increasing the  return-to-govalue or using the average from the offline dataset has little impact on the performance of multi-agent policy transfer. This demonstrates that our method is relatively stable and does not rely on precise tuning of return-to-go values.
>
> [1] Chen L, Lu K, Rajeswaran A, et al. Decision transformer: Reinforcement learning via sequence modeling[J]. Advances in neural information processing systems, 2021, 34: 15084-15097.
>
> [2] Xu M, Shen Y, Zhang S, et al. Prompting decision transformer for few-shot policy generalization[C]//international conference on machine learning. PMLR, 2022: 24631-24645.
>
> [3] Mao H, Zhao R, Chen H, et al. Transformer in transformer as backbone for deep reinforcement learning[J]. arXiv preprint arXiv:2212.14538, 2022.
>
>
> Your comments have greatly contributed to the improvement of our paper. If you have any new comments, please feel free to provide them, and we will promptly address them. If you find that we have addressed your concerns, we hope you will improve your rating.

---

> > ### Comment · Area_Chair_jgVf · 2025-08-06
> >
> > Dear reviewer cecZ,
> >
> > Could you please respond to authors' rebuttals as soon as possible?
> >
> > Thank you!
> > AC

---

> > ### Comment · Reviewer_cecZ · 2025-08-07
> > **I confirm my score**
> >
> > Thank you for the thoughtful and detailed rebuttal. I appreciate the clarifications regarding both the conceptual framing of the Bi-level Knowledge Transfer (BiKT) framework and the specifics around return-to-go initialization.
> >
> > Your explanation helped clarify the intent and novelty of the bi-level decomposition, particularly the explicit separation between tactic-level and skill-level transfer in multi-agent reinforcement learning. While the method builds on existing components, your framing introduces a meaningful and well-motivated abstraction that captures coordination as a first-class citizen in policy reuse. I now better appreciate the contribution as a novel framework for structuring knowledge transfer in MARL, which may inspire further research beyond the current implementation.
> >
> > I also found your explanation of the return-to-go setting and its consistency across environments helpful. The ablation you mentioned (on alternative return values) adds further confidence that the design choices are reasonable and robust.
> >
> > Overall, the rebuttal strengthens my initial positive impression. I remain confident in my recommendation to accept the paper and believe it makes a valuable contribution to the MARL and offline RL communities.

---

> > > ### Author Response · Authors · 2025-08-07
> > >
> > > Dear Reviewer,
> > >
> > > Thank you very much for taking the time to review our manuscript again. We sincerely appreciate your constructive feedback and thoughtful comments. We are pleased that we were able to address your concerns, and we are grateful for your support throughout the review process. Your review helped us recognize the importance of clearly highlighting our BiKT framework and clarifying the details of return-to-go, both of which have greatly improved the quality of our work.
> > >
> > > Thank you again for your valuable assistance and encouragement.
> > >
> > > Best regards,
> > >
> > > All authors

---

### Official Review · Reviewer_KvJL · 2025-06-28

**Clarity:** 3
**Significance:** 2
**Originality:** 2
**Rating:** 4
**Confidence:** 5

**Summary:**

This paper introduces a novel approach for enabling zero-shot generalization in multi-task MARL using offline data. The key insight is that effective policy transfer requires capturing knowledge at two levels: **individual agent skills** and **team-level cooperative tactics**. The authors consider scenarios where agents are trained on known tasks using offline data and must generalize to unseen tasks without further training. While prior work focuses solely on transferring individual agent skills, the authors argue this is insufficient for MARL. They introduce the concept of "tactics". **_Individual Skill Learning_** uses a VAE to extract transferable skill embeddings from offline trajectories that capture individual agent behaviors. **_Cooperative Tactic Learning_** employs a VQ-VAE to map skill combinations into a discrete tactic codebook. **_Bi-Level Decision Transformer_** is a policy model that first selects an appropriate tactic from the codebook, then infers individual skills conditioned on the tactic, and finally decodes actions.

**Questions:**

Please refer to the "Weaknesses" section for clarification questions.

1. How does the method ensure discovered tactics are semantically meaningful rather than arbitrary clusters? The regularization term $\beta_2 \sum_{k_1 \neq k_2} \frac{1}{||c_{k_1} - c_{k_2}||_2^2 + \epsilon}$ encourages diversity but not necessarily meaningfulness. What makes a skill combination constitute a "tactic" beyond being a cluster in embedding space? How do you ensure discovered tactics are semantically meaningful?
2. The choice of $L=10$ for the Decision Transformer seems arbitrary. How does performance vary with trajectory length?
3. How does BiKT perform when source and target tasks have fundamentally different objectives or constraints? The current experiments only vary team sizes within similar game mechanics.
4. What are the computational costs of the three-stage training? How does the method scale with team size, and what is the largest team size you've tested?
5. Why did you choose the original SMAC benchmark instead of SMACv2 [1], which was specifically designed to address multi-task generalization challenges in MARL? SMACv2 includes procedurally generated scenarios, randomized start positions, and more diverse unit compositions that would better test the generalization capabilities of BiKT.

[1] Ellis, Benjamin, et al. "SMACv2: An improved benchmark for cooperative multi-agent reinforcement learning." arXiv preprint arXiv:2212.07489 (2022).

**Ethical Concerns:**

["NO or VERY MINOR ethics concerns only"]

**Final Justification:**

After carefully considering all the points raised in the authors' response, I've decided to raise my score accordingly. I believe this final score accurately reflects the current state of the paper and its contributions. I do however believe it remains weak in its current form. I encourage the authors to continue their work in this direction, as the research area remains important and promising.

**Limitations:**

The paper's evaluation relies on the original SMAC benchmark, which may not adequately test the claimed generalization capabilities. SMACv2 [1] provides a more rigorous testbed for transfer learning in MARL due to several key advantages: procedural variation, dynamic opponent policies, compositional diversity. Given that the paper's central contribution concerns zero-shot generalization to "unseen tasks," the choice of SMAC potentially limits the strength of the empirical claims. The relatively constrained variation in SMAC scenarios may overestimate the method's ability to handle genuinely novel tactical situations that would arise in more diverse multi-agent settings.

**Recommendation**: The authors should either (1) include SMACv2 experiments to strengthen their generalization claims, or (2) explicitly acknowledge this limitation and discuss how SMAC's structural constraints may impact the generalizability of their findings to more varied scenarios.

[1] Ellis, Benjamin, et al. "SMACv2: An improved benchmark for cooperative multi-agent reinforcement learning." arXiv preprint arXiv:2212.07489 (2022).

**Paper Formatting Concerns:**

- Incomplete sentence Appendix B.1 [Offline Dataset] (L. 33) "The Medium policy is trained until it achieves approximately a 50..."
- Missing figure reference in Appendix D.3 (L. 138)

**Quality:**

3

**Strengths And Weaknesses:**

**Strengths**: The paper presents a well-formalized approach that properly grounds the bi-level knowledge transfer in established frameworks (VAE for skills, VQ-VAE for tactics, Decision Transformer for policy). The evaluation covers multiple task sets (Marine-Hard, Marine-Easy, Stalker-Zealot, Cooperative Navigation) with different data quality levels (Expert, Medium, Medium-Expert, Medium-Replay). The explicit separation of individual skills and team tactics is an original contribution that addresses a genuine gap in existing work. The bi-level knowledge decomposition provides a new lens for understanding multi-agent coordination that could influence future MARL research. The use of VQ-VAE to create a discrete tactic codebook elegantly handles the challenge of varying team sizes across tasks. The Bi-Level Decision Transformer that sequentially infers tactics then skills is a novel architectural contribution. The method successfully discovers meaningful skills and tactics that transfer across tasks, with visualizations showing clear clustering of learned representations. The paper makes a compelling case that multi-agent coordination knowledge exists at multiple levels of abstraction, and capturing both individual and team-level patterns is crucial for effective generalization in MARL.

**Weaknesses**: The paper lacks theoretical guarantees or analysis of when and why the bi-level decomposition should work. What conditions ensure that tactics are transferable across tasks? While ablations show robustness to some choices, critical hyperparameters like skill dimension ($N_s=4$) and codebook size ($K=16$) appear to be set arbitrarily without systematic justification. The three-stage training process (skills → tactics → policy) seems computationally expensive and potentially unstable. No analysis of computational costs or training time is provided. The distinction between skills and tactics, while intuitive, lacks formal mathematical definition. What exactly makes a skill combination a "tactic"? Key details are relegated to appendix without clear pointers. For instance, how exactly are entity features extracted and processed? Experiments are confined to relatively homogeneous task sets (same unit types, similar objectives). How well would this transfer to truly diverse tasks? All experiments involve relatively small teams (≤15 agents). The approach's scalability to larger multi-agent systems remains unclear. The baselines UPDeT and ODIS have already been outperformed; a comparison with their successors (e.g. TransfQMix [1], TETQmix [2], LDSA [3], RODE [4]), would strengthen claims. While the combination is novel, individual components (VAE, VQ-VAE, Decision Transformer) are well-established; the technical novelty is primarily in their integration. The paper doesn't deeply explore what makes a tactic transferable or how to ensure discovered tactics are meaningful beyond empirical observation.

[1] Gallici, M. et al. (2023) "TransfQMix: Transformers for Leveraging the Graph Structure of Multi-Agent Reinforcement Learning Problems", arxiv:2301.05334.

[2] Zhu, Y. et al. (2025) "Multi-task multi-agent reinforcement learning with task-entity transformers and value decomposition training", doi:10.1109/tase.2024.3501580.

[3] Yang, M. et al. (2022) "LDSA: Learning dynamic Subtask assignment in cooperative Multi-Agent Reinforcement Learning", arxiv:2205.02561.

[4] Wang, T. et al. (2022) "RODE: Learning Roles to Decompose Multi-Agent Tasks", arxiv:2010.01523

---

> ### Author Rebuttal · Authors · 2025-07-30
>
> We sincerely appreciate your recognition of our methodology, experimental results, and writing, along with the constructive feedback. Below, we offer detailed responses and clarifications. We hope these can resolve your concerns and enhance the strengths of our work.
> ## Main Questions
> > W3: The three-stage training process seems computationally expensive...
> >
> > Q4: What are the computational costs of the three-stage training?
>
> As reviewer Pgpu has kindly acknowledged as one of our strengths, **we have provided a detailed description of our computing resources in Appendix B.3.** Additionally, we report the training time for each method as follows:
>
> |  | UPDeT | ODIS | HiSSD | BiKT |
> | -| -| - | - | - |
> | Training Time (hours) per task |7|12|15|8|
>
> Compared to other methods, our three-stage training does not introduce additional computing resource consumption.
>
> > W4:  The distinction between skills and tactics lacks formal mathematical definition...
>
> We would clarify the relationship among three key variables: individual skills $z$, skill combination $S_c$, and tactic $c$.
>
> The skill combintaion $S_c$ is defined as the *Cartesian Product* of individual skills $z^1,...,z^n$, i.e., $S_c = z^1 \times z^2 \times...\times z^n$. This skill combination can serve as a representation of the team's overall behavioral pattern. However, due to variations in the number of agents across different tasks, the dimension of the skill combination differs even if the underlying team behavior from different environment maps remains the same. To address this challenge, we define the tactic as a mapping from skill combination: $\text{Mapping Func}(S_c=z^2 \times...\times z^n) \rightarrow c$
>  The $\text{Mapping Func}$ refers to our *Cooperative Tactic Encoder* $p_{tac}$ and *NNS* process in Figure 2.
>
> > W5:  Key details are relegated to appendix without clear pointers. For instance, how exactly are entity features extracted...
>
> In Figure 2, we mention that the details of the encoders and decoders are provided in Appendix C, where we also explain how our entity features are processed.  However, this may lead to misunderstanding as it does not explicitly clarify it. In the revision, we will modify it as "The network details and feature processing details about the encoders and decoders are shown in Appendix C". Besides, we will add some key feature process summary in sections 3.1 and 3.2.
> > W7: All experiments involve relatively small teams (≤15 agents)...
> >
> > Q4: How does the method scale ...
>
> For fair comparison, we adopt the same task setting and offline dataset as ODIS [1]. Beyond that, we also attempt an extended experiment with a larger number of agents. Due to time constraints, we will conduct experiments on additional maps in the future.
>
> | | ODIS | HiSSD | BiKT  |
> |-|-|-|-|
> |27m_vs_28m|$18.75 \pm 7.8$|$25.8 \pm 6.9$|$34.4 \pm 7.2$|
> > W8: ...a comparison with their successors (e.g. TransfQMix [1], TETQmix [2], LDSA [3], RODE [4]), would strengthen claims.
>
> These methods are not directly comparable to ours, as their problem settings differ fundamentally. Our problem setting focuses on **learning policies for multiple tasks simultaneously** from an **offline dataset**, with the goal of enabling **policy transfer to unseen tasks**. However, they do not conform to this setting and thus cannot serve as fair baselines for comparison.
>
> || Learning by online or offline dataset | Can learn multiple tasks simultaneously | Can policy transfer |
> |-|-|-|-|
> |TransfQMix|online|No| Yes|
> |TETQmix|online|Yes| Yes|
> |LDSA|online|No| No|
> |RODE|online|No|Yes|
> |BiKT|offline dataset|Yes| Yes|
>
> > W9:  While the combination is novel, individual components (VAE, VQ-VAE, Decision Transformer) are well-established; the technical novelty is primarily in their integration.
>
> Our primary purpose is to validate the effectiveness of our BikT paradigm by components. For further details, please refer to our Response 1 to *Reviewer cecZ*.
>
> > W10: The paper doesn't deeply explore what makes a tactic transferable...
> >
> >Q1: How does the method ensure discovered tactics are semantically meaningful...
>
> * **Difference between arbitrary clustering of skill combinations and learned tactic.**
>   A key limitation of clustering based on skill combinations is that their dimensionality depends on the number of agents, making it challenging to generalize across tasks with varying agent counts. Our tactic overcomes this by mapping skill combinations into a fixed-dimensional space.
> * **Meaningfulness of the tactics.**
>   Figures 4(c) and 4(d) illustrate the distribution of tactics across different tasks, revealing both distinctions and similarities. For instance, tasks like 3m and 5m6m exhibit distinct tactic distributions, whereas 2s3z and 2s4z show similar team-level tactical patterns. This indicates that the learned tactics offer task-specific team guidance, rather than forcing a single multi-agent policy to generalize without considering task-specific preferences.
>
> > W6: Experiments are confined to relatively homogeneous task sets...
> >
> > Q3: How does BiKT perform when source and target tasks have fundamentally different objectives or constraints?
> >
> > Q5: Why did you choose the original SMAC benchmark instead of SMACv2...
>
> We choose SMAC as the benchmark to ensure a fair comparison, as both our dataset and task setup follow ODIS [1].
>  However, we also believe it is meaningful to evaluate our method in smac v2. To this end, we train QMIX for 5 million steps on protoss_3_3 and protoss_5_5, collect 2,000 trajectories from the trained policies, and then conduct experiments using this offline data. we can conclude that BiKT still achieves high performance in more dynamic environment.
> | | (source) protoss_3_3| (source) protoss_5-5 | (unseen) protoss_4_4|(unseen) protoss_6_6|(unseen) protoss_7_7 |(unseen) protoss_8_8|
> |-|-|-|-|-|-|-|
> |mean winning rate of offline data | $63.5$| $54.4$ |-|-|-|-|
> |ODIS| $43.3 \pm 4.7$| $21.8 \pm 5.4$| $18.8 \pm 7.7$| $20.8 \pm 7.6$| $15.6 \pm 8.1$|$3.1 \pm 1.5$|
> |HiSSD| $47.6 \pm 6.7$| $38.8 \pm 6.1$| $38.6 \pm 5.9$| $26.7 \pm 6.3$ | $21.9 \pm 7.8$| $5.2 \pm 3.9$|
> |BiKT| $54.9 \pm 5.4$|$41.4 \pm 7.3$|$35.4 \pm 9.8$|$41.5 \pm 8.7$| $20.8 \pm 6.9$|$15.7 \pm 7.4$ |
>
> > Format: Incomplete sentence Appendix B.1...; Missing figure reference...
> We will modify it as "... it achieves approximately a 50 winning rate." And we will add the figure reference.
>
> ## Ablation Study
> |Different settings|source 3m| source 5m6m| source 9m10m| unseen4m | unseen5m | unseen10m | unseen12m | unseen 7m8m | unseen 8m9m | unseen 10m11m|unseen10m12m | unseen13m15m |
> |-|-|-|-|-|-|-|-|-|-|-|-|-|
> | w/o Tactic Codebook $C$ | $98.2 \pm 0.2$  | $68.2 \pm 5.3$ | $83.2 \pm 2.5$ | $90.2 \pm 1.9$ | $88.6 \pm 2.7$ | $86.2 \pm 3.2$ | $57.2 \pm 4.2$ | $23.3 \pm 5.5$  | $20.3 \pm 9.2$ | $40.7 \pm 9.2$ | $1.6 \pm 1.6$ | $1.6 \pm 1.3$ |
> | $L=5$   | $100.0 \pm 0.0$ | $78.9 \pm 3.5$ | $98.4 \pm 1.5$ | $99.3 \pm 0.9$ | $99.9 \pm 0.8$  | $97.2 \pm 1.5$ | $98.7 \pm 0.9$ | $64.8 \pm 8.2$ | $48.2 \pm 8.3$ | $90.2 \pm 2.3$ | $12.0 \pm 1.8$ | $3.2 \pm1.5$ |
> | $N_s=10$ | $100.0 \pm 0.0$ | $81.8 \pm 3.6$ | $99.2 \pm 1.5$ | $98.8 \pm 0.2$ | $100.0 \pm 0.0$ | $99.4 \pm 0.1$ | $98.8 \pm 1.2$| $70.3 \pm 10.3$| $47.5 \pm 8.4$ | $89.4 \pm 4.0$ | $13.4 \pm 3.8$ | $4.1 \pm0.3$ |
> | $K=32$ | $100.0 \pm 0.0$ | $80.9 \pm 3.2$ | $99.2 \pm 0.3$ | $99.2 \pm 0.1$ | $99.3 \pm 0.2$ | $99.3 \pm 0.2$ | $99.0 \pm 0.2$ | $70.2 \pm 8.0$ | $46.2 \pm 7.4$ | $90.2 \pm 2.1$ | $12.3 \pm 2.2$ | $3.7\pm 1.6$ |
> | BiKT  | $100.0 \pm 0.0$ | $81.3 \pm 4.5$ | $99.4 \pm 0.4$ | $99.3 \pm 0.1$ | $100.0 \pm 0.0$ | $99.4 \pm 0.1$ | $99.0 \pm 0.2$ | $68.0 \pm 9.9$ | $50.0 \pm 6.2$ | $90.6 \pm 1.1$ | $14.6 \pm 1.5$ |$4.2 \pm 2.1$ |
>
> > W1: The paper lacks theoretical guarantees or analysis of when and why the bi-level decomposition should work...
>
> This question is both insightful, and the answer to it directly highlights our fundamental advantages. In the paper, we have indeed considered this point and conducted ablation studies. We consider 2 settings:
> * Remove the tactic learning. **(We have conducted it in line 879 of the Appendix.)** We copy the results in the above table.
> * Remove the tactic learning and individual skill learning. **(We have conducted it in line 867 of the Appendix.)**
>
> > W2: Critical hyperparameters like skill dimension ($N_s=4$) and codebook size ($K=16$) appear to be set arbitrarily...
>
> We consider the following setting:
> * codebook size $K=32$ (**We have conducted it  in line 873 of the Appendix, we copy the result in the above table.**)
> * Skill dimension $N_s=10$
>
> > Q2: The choice of $L=10$ for the DT seems arbitrary...
>
> We consider the following setting:
> * Content length $L=5$ (**We have conducted it in line 905-909, we copy the results here.**)
>
> **Summary:**
> * **Bi-level transfer matters**: **Tactics provide team-level guidance across tasks.**  Without them, the policy struggles in multitask transfer, relying solely on observation changes of different tasks. **Individual skills help generalize to varying agent numbers by learning action mappings of different lengths.** Without both, the method only generalizes to simple tasks like 3m to 5m.
> * **The perturbation of skill dimension has little impact on BiKT’s performance.**
> * **Effective tactics in the codebook converge** regardless of the codebook size (see Lines 908–911).
> * **BDT doesn't rely on the model scale advantage brought by longer DT content length.** We didn't use arbitrary $L$ but follow the common practice in Decision Transformer ($L=5$). We double it to 10 and the performance remains consistent.
>
> [1] Fuxiang Zhang, et.al. Discovering generalizable multi-agent coordination skills from multi-task offline data. In ICLR 2022
>
> Thank you for your helpful comments and they’ve significantly improved our paper. If you have further suggestions, we’d be happy to address them. If you feel your concerns have been resolved, we kindly hope you’ll improve your rating.

---

> > ### Comment · Area_Chair_jgVf · 2025-08-06
> >
> > Dear reviewer KvJL,
> >
> > Could you please respond to the authors' rebuttals as soon as possible?
> >
> > Thank you!
> > AC

---

> > ### Comment · Reviewer_KvJL · 2025-08-06
> >
> > I thank the authors for their thorough and detailed response to my review. The additional clarifications and explanations provided have been helpful in addressing several of my underlying questions and concerns. After carefully considering all the points raised in the authors' response, I have reassessed the paper in light of these clarifications. I appreciate the authors' efforts to address the feedback and the additional context provided, it warrants an increase in rating which will accurately reflect the current state of the paper and its contributions.

---

> > > ### Author Response · Authors · 2025-08-07
> > >
> > > Dear Reviewer,
> > >
> > > Thank you very much for your valuable time and for reviewing our manuscript again. We sincerely appreciate your constructive feedback and the improved rating. Your insightful comments have greatly helped us enhance the quality of our work. We are pleased that we were able to address your concerns, and we are grateful for your support throughout the review process.
> > >
> > > Thank you again for your assistance and encouragement.
> > >
> > > Best regards,
> > > All authors

---

### Official Review · Reviewer_Pgpu · 2025-07-02

**Clarity:** 2
**Significance:** 3
**Originality:** 3
**Rating:** 5
**Confidence:** 3

**Summary:**

This paper provides a method for skill level and tactic level transfer in the Multi-agent Multi-task Reinforcement Learning setting. Skills are low-level agent-specific abilities that are combined in tactics which are employed by multiple agents in a cooperative setting. The proposed methods incorporates 3 levels of learning to learn the key components, which include the action decoder, tactic codebook, and (Bi-level) Decision Transformer. Experiments are carried out on StarCraft 2 Micromanagement and Cooperative Navigation environments against several relevant baselines.

**Questions:**

# Questions
- Line 104: The subscript $t$ is missing from the policy definition. Is this on purpose? I.e., shouldn't it be $\pi^i(a^i_t | \tau^i_t)$?
- Line 114: What is the policy pre-trained on and why is this policy pre-trained. Why not use a random policy?
- Figure 2: What do you mean by "feature of entity $i$", what exactly is the feature?
- Section 4.1: Am I correct in understanding that the individual skills that are learned are from trajectories that are generated from policies that already solve a particular task? If so, what is the reasoning for this?
- What is the stop-gradient operator used for in Equation 4?

**Ethical Concerns:**

["NO or VERY MINOR ethics concerns only"]

**Final Justification:**

I think that the authors have addressed the questions and concerns that I had, and have also used the suggestions I had made for clarity, etc. Thus I have increased my score to 5.

**Limitations:**

Yes

**Quality:**

4

**Strengths And Weaknesses:**

## Strengths
- I think that the idea is an interesting one. Furthermore, the diagram were good.
- The experimentation seems good with a wide variety of baselines being compared to.
- I appreciate the ablation studies. I understand that page limits caused them to be moved to the Appendix. Could a summary of what was carried out and the key outcomes be added to the main text however?
- I also appreciate that computing resources used are listed in Appendix B.3. However, I would be interested in knowing how long (assuming that the same compute resources) each of the methods in Table 1 and 2 took.
## Weaknesses
- Please provide citations for SMAC and MPE from the beginning of the paper rather than starting from Section 5.
- There are some grammatical and notation errors and inconsistencies. This generally made the paper difficult to read. Please see below.
#### Notation and writing style
- Section 3.1: I think the authors should be careful about what are spaces and elements of the spaces. E.g., in line 97, I think $s$ and $o$ are the state *space* and observation *space* respectively and so should be upper-case and defined as such (in line 98).
- Line 97, line 115, line 163: Please be consistent with whether you use "$\langle\rangle$", "$()$" or $[]$ for defining a tuple. I would prefer using $()$.
- Line 104: The notion for the policy doesn't follow the appropriate notation for a function signature. It is correctly written for the transition function and the reward function. Either use a similar notation, or writing something like $\pi^i(a^i | \tau^i) \in [0, 1]$.
- Line 131 "... to as individual skills $z$; and (2) The ...": I think replace the semi-colon with a comma and make "The" lowercase (to "the"). The same should be done on line 139 and line 140.
- Line 137 "In specific...": I think remove "In specific" or replace with "Specifically".
- Line 150 "...$\mathbb{R}^{N_s}$ is the dimension of the skill embedding": Did you mean that "$N_s$ is the dimension of the skill embedding"?
- Line 160 "While single-agent...": Did you mean "While *a* single agent..."?
- Line 150: What is $\hat{a}$ and $\hat{z}$?
- Equation 3, 4: The input parameters $\phi_1$, etc don't appear on the right-hand sides of the equations.
- Line 163: I think $[z^1 \times z^2 \times ... \times z^N]$ should be $(z^1, z^2, ..., z^N) \in \mathbb{R}^{N_s} \times \mathbb{R}^{N_s} \times ... \times \mathbb{R}^{N_s}$. As mentioned before, please use *spaces* and *elements of spaces* correctly.
### Suggestions:
- Section 3.1 and 3.2: I think it might be worth highlighting that the superscript $i$ refers to different agents whereas the superscript $n$ refers to different source training tasks.

---

> ### Author Rebuttal · Authors · 2025-07-30
>
> Thank you sincerely for your recognition of our methodology, experimental results, and writing, as well as for your thoughtful and constructive feedback. In response to your comments, we would like to offer the following clarifications. We hope these explanations help address your concerns and further highlight the strengths of our work.
>
> ## Main Questions
>
> > Strength 4: I also appreciate that computing resources used are listed in Appendix B.3. However, I would be interested in knowing how long (assuming that the same compute resources) each of the methods in Table 1 and 2 took.
>
>   Under our computing resource, the whole training time for each method is as follows:
>
> |                                | UPDeT | ODIS | HiSSD | BiKT |
> | ------------------------------ | ----- | ---- | ----- | ---- |
> | Training Time (hours) per task | 7     | 12   | 15    | 8    |
>
> > W9: What is $\hat{a}$ and $\hat{z}$?
>
>   We apologize for not clearly explaining this in the paper. $\hat{a}$ refers to the output of the Action Decoder $q_{act}$, and $\hat{z}$ refers to the output of the Skill Decoder $q_{skill}$.
>   We use $\hat{a}$ and $\hat{z}$ to distinguish the decoded outputs from the original action $a$ and the learned skill embedding $z$ in the trajectory.
> > Q2: Line 114: What is the policy pre-trained on and why is this policy pre-trained. Why not use a random policy?
> >
> > Q4: Section 4.1: Am I correct in understanding that the individual skills that are learned are from trajectories...
>
> * Why not use a random policy?
>
>   Offline reinforcement learning (Offline RL) refers to the paradigm where agents learn solely from a fixed dataset of past interactions without interacting with the environment [1].
>
>   As noted in [2, 3], offline RL critically depends on datasets containing expert-level trajectories. In contrast, datasets collected using random policies are typically of low quality, often containing a large number of uninformative or low-reward actions. It leads to a fundamental challenge: the agent is trained on suboptimal behavior data but expected to perform expert-level decisions at test time, which often results in severe performance degradation [2].
>
> * Why is the policy pre-trained? Why use a pretrained policy to collect trajectories?
>
>     In fact, the trajectories in the offline dataset exhibit expert-level behavior, aiming for the agent to learn an optimal policy [2]. Such offline RL datasets can be collected either via human demonstrators,  rule-based policies, or through pre-trained methods [3, 4], like QMIX in MARL.
>
>   It is worth noting that QMIX requires separate online training and trajectory collection for each individual task. Due to the differences in state and action spaces across tasks, QMIX cannot be directly applied to multiple tasks, nor can it support offline learning.
> * What is the policy pre-trained on？
>
>   For a fair comparison, all methods are evaluated using the same offline datasets, which are provided by ODIS [5] and collected using a pre-trained QMIX policy. Specifically, each offline dataset of tasks contains trajectories of varying quality levels: *Expert*, *Medium*, *Medium-Expert*, and *Medium-Replay*, as detailed in Table 6 and Table 7. These datasets are collected by controlling the win rate thresholds of QMIX.
>
>   * The *Expert* policy is trained for 2,000,000 environment steps.
>   *  The *Medium* policy is trained until it reaches approximately a 50% win rate.
>   * The *Medium-Expert* dataset is a combination of trajectories from both *Expert* and *Medium* policies.
>   * The *Medium-Replay* dataset is collected from the replay buffer of the *Medium* policy, which contains a higher proportion of low-quality trajectories.
>
>   We have considered expert trajectories of varying quality across different datasets and conducted experiments to validate the effectiveness of our method.
>
> > Q3: Figure 2: What do you mean by "feature of entity $i$ "...
>
> The entity feature is a concept distinct from observation, and can be seen as a subset of it. The feature of entity $i$ represents the information of agent $i$ (e.g., position, health), while the observation of agent $i$ refers to the information of multiple entities within its observation range. In contrast, the state contains the complete set of entity features for all entities.
>
> The dimensionality of an agent's observation can vary across different tasks, but the dimensionality of entity features remains fixed. So we use entity features as the input to the encoder to better support multi-task learning.
>
> > Q5: What is the stop-gradient operator used for in Equation 4?
>
> We use the default stop-gradient operator as the original VQ-VAE method [6], which is essential for enabling end-to-end training despite the non-differentiable quantization step [6]. The outputs of *Cooperative Tactic Encoder* $p_{tac}$ encoder $\hat{c}_t$ are quantized to the nearest embedding vector $c^k$ from the Tactic codebook $C$. However, the quantization step is non-differentiable, so to enable end-to-end training, the stop gradient is set as in Equation 9. That is, in the forward pass, it evaluates to the same value, but during the backward pass, no gradients are propagated.
>
> ## Notation and writing style
>  As you mentioned, there is still room for improvement in terms of notation and writing.  We apologize for any inconvenience for reading and appreciate your advice.  Your comments are relevant and helpful. In the revised version, we will refine the logical flow and improve the clarity of mathematical notation. Below, we address your concerns one by one.
> > Strengths 3: I appreciate the ablation studies. ...  Could a summary of what was carried out and the key outcomes be added to the main text however?
>
> We agree with this point. In the revision, we plan to reorganize the manuscript and add a summary of the ablation study around Section 5.2, line 260.
> > W1: Please provide citations for SMAC and MPE from the beginning of the paper...
>
> In the revision, we will provide citations for SMAC and MPE at line 64.
> > W2:  Section 3.1: I think the authors should be careful about what are spaces and elements of the spaces;
> >
> > W10: I think$[ z^1 \times z^2 \times ... \times z^N]$ should be...
>
> In the revision, we will clarify the distinction between spaces and the elements of spaces. Specifically, we will change *s* and *o* to uppercase in line 97. In line 163, in fact, the skill combination is the Cartesian product of skills $(z^1 , z^2 , ...)$. For the sake of standardization, we will moidfy $[ z^1 \times z^2 \times ... \times z^N]$ as $(z^1 , z^2 , ... , z^N) \in \mathbb{R}^{N_s} \times \mathbb{R}^{N_s} \times ... \times \mathbb{R}^{N_s}$
> > W3: Line 97, line115,  line 163: Please be consistent with whether you use "$<>$", "$()$" or$[]$ for defining a tuple.
>
> In the revision, we will uniformly use "()" for defining a tuple.
>
> > W4: The notion for the policy doesn't follow the appropriate notation for a function signature....; Q1：Line 104: The subscript is missing from the policy definition. Is this on purpose?..
>
> In the revision, we will modify $\pi^i(a^i | \tau^i) \rightarrow [0, 1]$ in line 104 as $\pi^i(a^i_t | \tau^i_t) \in [0, 1]$.
>
> > W5:  ...I think replace the semi-colon with a comma and make "The" lowercase (to "the")...
>
> In the revision, we will modify "The" to "the" in lines 139 and 140.
>
> > W6: Line 137 "In specific...": I think remove "In specific" or replace with "Specifically"
>
> We agree that removing the “In specific...” enhances readability. In the revision, we will modify it.
>
> > W7: Line 150 , ... Did you mean that $N_s$ is the dimension of the skill embedding"?
>
> Yes, the $N_s$ is the dimension of the skill embedding; we will modify it in the revision.
>
> > W8: Line 160 "While single-agent...": Did you mean "While *a* single agent..."?
>
> Yes, for standardization, we will modify it as "a single agent" in the revision.
>
> > Suggestion 1: Section 3.1 and 3.2: I think it might be worth highlighting that the superscript $i$ refers to different agents whereas the superscript $n$ refers to different source training tasks.
>
> Thank you for pointing this out. In fact, we have also been considering how to help readers quickly distinguish between the symbols for agents and tasks. Placing the clarification here indeed makes the distinction clearer from the beginning. We will emphasize the difference in notation between agents and tasks at the end of Section 3.2 to improve the clarity of the paper.
>
> [1] Levine S, Kumar A, Tucker G, et al. Offline reinforcement learning: Tutorial, review, and perspectives on open problems[J]. arXiv preprint arXiv:2005.01643, 2020.
>
> [2] Fujimoto S, Meger D, Precup D. Off-policy deep reinforcement learning without exploration[C]//International conference on machine learning. PMLR, 2019: 2052-2062.
>
> [3] Agarwal R, Schwarzer M, Castro P S, et al. Deep reinforcement learning at the edge of the statistical precipice[J]. Advances in neural information processing systems, 2021, 34: 29304-29320.
>
> [4] Fu J, Kumar A, Nachum O, et al. D4rl: Datasets for deep data-driven reinforcement learning[J]. arXiv preprint arXiv:2004.07219, 2020.
>
> [5] Fuxiang Zhang, Chengxing Jia, Yi-Chen Li, Lei Yuan, Yang Yu, and Zongzhang Zhang. Discovering generalizable multi-agent coordination skills from multi-task offline data. In The Eleventh International Conference on Learning Representations, 2022.
>
> [6] Van Den Oord A, Vinyals O. Neural discrete representation learning[J]. Advances in neural information processing systems, 2017, 30.
>
> Thanks again for your conscientious review. Your comments have been immensely helpful in improving our paper. If you have any additional feedback, please don’t hesitate to share it, and we will address it promptly. If you feel that our responses have adequately resolved your concerns, we hope you will consider improving your rating.

---

> > ### Comment · Area_Chair_jgVf · 2025-08-06
> >
> > Dear reviewer Pgpu,
> >
> > Could you please take time to respond to authors' rebuttals as soon as possible?
> >
> > Thank you!
> > AC

---

> > ### Comment · Reviewer_Pgpu · 2025-08-06
> > **Response to rebuttal**
> >
> > I thank the authors for the detailed response to my questions and suggestions. I believe my concerns are addressed and I will increase my score accordingly!

---

> > > ### Author Response · Authors · 2025-08-07
> > >
> > > Dear Reviewer,
> > >
> > > Thank you for taking the time to review our paper again and increasing the score. We really appreciate your helpful comments and the chance to make our work better based on your suggestions. We are happy that we were able to address your concerns, and we are thankful for your useful feedback during the review process.
> > >
> > > Thank you again for your support and for helping us make our work better.
> > >
> > > Best regards,
> > >
> > > All authors

---

### Official Review · Reviewer_9d35 · 2025-07-02

**Clarity:** 3
**Significance:** 2
**Originality:** 2
**Rating:** 5
**Confidence:** 1

**Summary:**

This paper introduces BiKT, a novel approach for knowledge transfer across tasks for MARL at both individual and team level. The algorithm results in better performances on SMAC and CN compared to baselines, indicating the effectiveness of the method. Visualization and ablation studies help with better understanding what the neural network is actually learning.

**Questions:**

See Weaknesses.

**Ethical Concerns:**

["NO or VERY MINOR ethics concerns only"]

**Final Justification:**

The rebuttal has addressed my concerns.

**Limitations:**

yes

**Quality:**

3

**Strengths And Weaknesses:**

Strengths:

1. The proposed approach shows improvements on well-established benchmarks, outperforming the baseline methods by a large margin on some tasks. Especially on unseen tasks.

2. The visualization of the learned skills help with comprehensive understanding of the method.

3. Extensive experimental results and analyses make the paper really solid.

Weaknesses:

1. If the learned skills can be clustered into a number of groups, can we also consider a codebook for learning it? i.e., use VQ-VAE as well for skill learning.

2. The intuition behind using codebook for team-level tactics and no codebook for skills is unclear. Have the authors tried different combinations of network formulation as an ablation study?

3. In Table 1, for some tasks like 10m11m in Medium-Expert, the proposed method significantly underperforms the baseline method, can the author provide some clarifications for this?

---

> ### Author Rebuttal · Authors · 2025-07-29
>
> Thank you very much for your recognition of our paper's methodologies, empirical results, paper writing, and the valuable feedback you provided. In response to your comments, we would like to make the following clarifications and feedback. We hope our explanations and analyses can eliminate concerns and make you find our work stronger.
>
> > Weakness1: If the learned skills can be clustered into a number of groups, can we also consider a codebook for learning it? i.e., use VQ-VAE as well for skill learning.
> >
> > Weakness2: The intuition behind using codebook for team-level tactics and no codebook for skills is unclear. Have the authors tried different combinations of network formulation as an ablation study?
>
> We adopt different strategies for learning tactic embeddings and skill embeddings, each with a distinct focus.
>
> For tactic embeddings, which serve as a shared team-level guiding signal for all agents, we represent them as fixed or discrete values. This design enables all agents to access the same team guidance, facilitating coordinated cooperation. Additionally, we employ VQ-VAE to learn a compact set of tactic embeddings across multiple MARL tasks, allowing the discovery of a limited set of reusable team strategies. In contrast, using a standard VAE for tactic learning would result in a unique team guidance vector for each trajectory in every task, introducing higher randomness and instability when generalizing to unseen tasks.
>
> For skill embeddings, we use a VAE to preserve the diversity and distributional structure of the skill embedding space. The agents' action semantics vary across environments and agent numbers. Even when two agents in different scenarios exhibit similar behaviors or skills, we still expect their skill embeddings to differ. Using VAE allows for these embeddings to be mapped to actions with different physical meanings in different environment maps.
>
> **In summary:**
>
> - **Why not use VQ-VAE for skill embeddings?**
>    While skill embeddings could be clustered, we avoid using VQ-VAE to retain their diversity and the rich distributional properties of the embedding space, which facilitates skill embeddings mapping to different actions to suit environments, thereby improving multi-task performance.
> - **Why not use VAE for tactic embeddings?**
>    To promote cooperation, we prefer agents to share a fixed team guidance. VQ-VAE helps extract a compact set of stable tactics, reducing variability and instability in the generalization process.
>
> In addition, we conducted the following ablation studies for validation:
>
> * **A1**: Replace the VAE with a VQ-VAE for skill embedding learning, and learn a skill embedding codebook with size 32.
>
> * **A2**: Use a VAE instead of a discrete embedding for tactic learning, followed by clustering into 16 categories. **We have conducted it in the Appendix**, lines 879-881. And the results are shown in Table 11. We list them here for convenience.
>
> * **A1_A2**: Apply both A1 and A2 simultaneously.
>
> | Different RTG | source-task 3m  | source-task 5m6m | source-task 9m10m | unseen-task 4m | unseen-task 5m  | unseen-task 10m | unseen-task 12m | unseen-task 7m8m | unseen-task 8m9m | unseen-task 10m11m | unseen-task 10m12m | unseen-task 13m15m |
> | ------------- | --------------- | ---------------- | ----------------- | -------------- | --------------- | --------------- | --------------- | ---------------- | ---------------- | ------------------ | ------------------ | ------------------ |
> | A1            | $90.6 \pm 4.4$  | $43.6 \pm 15.6$  | $53.1 \pm 15.3$   | $93.8 \pm 2.2$ | $88.4 \pm 8.1$  | $60.3 \pm 10.3$ | $53.1 \pm 9.7$  | $13.5 \pm 12.8$  | $9.8 \pm 6.5$    | $28.1 \pm 7.2$     | $0 \pm 0$          | $0 \pm 0$          |
> | A2            | $99.2 \pm 0.4$  | $64.3 \pm 6.7$   | $80.3 \pm 4.4$    | $92.2 \pm 2.4$ | $90.6 \pm 4.6$  | $88.3\pm 4.2$   | $60.5 \pm 3.8$  | $18.2 \pm 7.4$   | $17.8 \pm 5.2$   | $38.7 \pm 8.3$     | $0.9 \pm 0.5$      | $0.5 \pm0.4$       |
> | A1_A2         | $87.5 \pm 5.4$  | $33.8 \pm 16.0$  | $51.9  \pm 17.9$  | $90.6 \pm 6.2$ | $84.4 \pm 9.7$  | $59.4 \pm 22.0$ | $43.8 \pm 11.8$ | $6.8 \pm 6.2$    | $6.25 \pm 3.7$   | $25.0 \pm 4.4$     | $0 \pm 0$          | $0 \pm 0$          |
> | BiKT          | $100.0 \pm 0.0$ | $81.3 \pm 4.5$   | $99.4 \pm 0.4$    | $99.3 \pm 0.1$ | $100.0 \pm 0.0$ | $99.4 \pm 0.1$  | $99.0 \pm 0.2$  | $68.0 \pm 9.9$   | $50.0 \pm 6.2$   | $90.6 \pm 1.1$     | $14.6 \pm 1.5$     | $4.2 \pm 2.1$      |
>
> From the above experimental results, we observe the following:
>
> * Learning skill embeddings as a codebook (A1) limits policy generalization, as it cannot adapt to action mappings with varying dimensions across different maps.
>
> * Using a VAE followed by clustering for tactic embeddings (A2) reduces stability during generalization.
>
> * When both A1 and A2 are applied simultaneously, overall performance degrades significantly.
>
> Thank you for pointing this out to us.  We believe that discussing the separate learning of skills and tactics is important, as it helps clarify the details of our bi-level knowledge framework. We will include this discussion in the revision.
>
> > Q1: In Table 1, for some tasks like 10m11m in Medium-Expert, the proposed method significantly underperforms the baseline method, can the author provide some clarifications for this?
>
> Thank you for pointing this out. We attribute the performance gap to a combination of two factors:
>
> * **Unbalanced quality of task trajectories.**
>   In non-expert datasets, different source tasks exhibit varying winning rates. For example, in the *Medium* offline dataset, the winning rate for source tasks *3m*, *5m_vs_6m*, and *9m_vs_10m* are 54.02%, 27.51% and 41.46%.  This imbalance introduces biases in the policy quality learned from different tasks, which in turn affects generalization performance on unseen tasks. The unbalanced quality of task trajectories is a common challenge faced by all methods. We consider it a shared issue— as the quality of the offline dataset declines, each method tends to develop biases toward certain tasks during learning, which in turn leads to increased instability in performance across tasks.
>
> * **Accumulated sub-optimality in the bi-level learning structure.**
>   The individual skill learning depends on the quality of the offline dataset, while tactic learning further relies on both the learned individual skills and the dataset quality. When the offline data quality drops from expert level, the individual skill learning captures suboptimal action mappings, which then propagate to the tactic learning stage, resulting in suboptimal team-level strategies. Such accumulation of sub-optimal strategies biases the learning process, preventing some tasks from achieving expert-level performance.
>
> However, experimental results of Table 1 show that when the quality of trajectories in the offline dataset degrades, our method maintains the highest level of stability, with the smallest performance drop. This indicates that despite the sub-optimal trajectories introduced by lower-quality offline data, our Bi-Level Knowledge strategy is able to preserve robust and stable policy learning.
>
> All your comments have greatly contributed to the improvement of our paper. If you have any new comments, please feel free to provide them, and we will promptly address them. If you find that we have addressed your concerns, we hope you will reconsider your rating.

---

> > ### Comment · Reviewer_9d35 · 2025-08-04
> >
> > Thanks the author for the clarification. All of my concerns has been addressed. I appreciate the ablation study of different model choices for tactics and skills, which provides insights when paired with the analysis the author provided. I'll increase my score by 1.

---

> > > ### Author Response · Authors · 2025-08-05
> > >
> > > Dear Reviewer,
> > >
> > > Thank you very much for taking the time to re-evaluate our submission and for raising the score. We truly appreciate your thoughtful feedback and the opportunity to improve our work based on your suggestions. We are glad that we addressed your concerns, and we remain grateful for your constructive comments throughout your review.
> > >
> > > Thank you again for your support and for helping us improve the quality of our work.
> > >
> > > Best regards,
> > >
> > > All authors

---

### Author Response · Authors · 2025-08-07

Dear Reviewers, ACs, SACs, and PCs,

We sincerely thank you for your hard work throughout the review and rebuttal process. We deeply appreciate the constructive and insightful comments provided by the reviewers after carefully reading our manuscript, which have greatly improved the quality of our work.

***

Here, we provide a summary of the entire rebuttal process:

* The motivation and novelty of our work were convincing, for example:
     * Reviewer  **Pgpu** said: "I think the idea is an interesting one"
     * Reviewer  **cecZ** said: "Clear motivation and novel framing"


* The clarity of our work was consistently appreciated by reviewers, for example:
     * Reviewer  **9d35** said: "The visualization of the learned skills helps with comprehensive understanding of the method."
     * Reviewer  **cecZ** said: "Clarity and structure: The writing is excellent, concise, easy to follow, and logically structured. Figures (especially Figure 1) are helpful and informative.
"

* Through the rebuttal process, we addressed most of  reviewers' concerns.
     * Reviewer **9d35** said: " All of my concerns have been addressed."
     * Reviewer **Pgpu** said:  "I believe my concerns are addressed."
     * Reviewer **KvJL** said: "...have been helpful in addressing several of my underlying questions and concerns."
     * Reviewer **cecZ** said: "Your explanation helped clarify the intent and novelty of the bi-level decomposition; I also found your explanation of the return-to-go setting and its consistency across environments helpful. "



* Based on the rebuttal process, we will further improve our paper as follows:

     * Following Reviewer **9d35**, we will add a discussion to emphasize why we use VAE for skill learning and VQVAE for tactic learning, and add a discussion about how decreased trajectory quality affects performance.
     * Following suggestions from Reviewers **Pgpu** and **KvJL**, we will further revise and standardize the writing for better clarity and readability.
     * Following Reviewer **KvJL**, we plan to extend future experiments to larger-scale settings (with more agents and more complex environments) in future work.
     * Following Reviewer **cecZ**, we will clarify the initialization of return-to-go and add the discussion to highlight the importance of bi-level knowledge transfer.


***

In the end, we would like to thank you again for your time and dedication to the review process. We appreciate that we receive many valuable feedback from the reviewers. We hope that our work will earn your further support. We fully respect and support all your future judgment on our paper. Thank you very much.

Best regards,

All authors

---

### Decision · Program_Chairs · 2025-09-17

**Decision:**

Accept (poster)

**Comment:**

This paper proposes a bi-level knowledge transfer framework for multi-task MARL, enabling transfer at both the individual skill level and the inter-agent coordination level. The method builds upon offline reinforcement learning with expert-level trajectories, incorporating a hierarchical structure and discrete encodings for skills and tactics.

The strengths of the paper lie in its well-motivated problem setting, the bi-level transfer strategy, and extensive empirical validation on multiple benchmarks (MPE and SMAC). Reviewers acknowledged the methodological soundness and empirical performance, and raised only minor concerns.

During the rebuttal, the authors provided thorough responses, clarifying key design decisions such as the use of expert trajectories, policy pre-training, and entity features. They also addressed writing and notation issues, promising improvements in the revised version.

Given the sound methodology, strong experimental results, and the authors’ detailed and constructive responses to all raised concerns, the paper appears to be in a good shape for acceptance.